# The *Plasmodium falciparum* apicoplast cysteine desulfurase provides sulfur for both iron-sulfur cluster assembly and tRNA modification

Russell P Swift[1,2†‡], Rubayet Elahi[1,2†], Krithika Rajaram[1,2], Hans B Liu[1,2], Sean T Prigge[1,2]*

[1]Department of Molecular Microbiology and Immunology, Johns Hopkins University, Baltimore, United States; [2]The Johns Hopkins Malaria Research Institute, Baltimore, United States

*For correspondence:
sprigge2@jhu.edu

†These authors contributed equally to this work

Present address: ‡Division of Biology and Biological Engineering, California Institute of Technology, Pasadena, United States

**Abstract** Iron-sulfur clusters (FeS) are ancient and ubiquitous protein cofactors that play fundamental roles in many aspects of cell biology. These cofactors cannot be scavenged or trafficked within a cell and thus must be synthesized in any subcellular compartment where they are required. We examined the FeS synthesis proteins found in the relict plastid organelle, called the apicoplast, of the human malaria parasite *Plasmodium falciparum*. Using a chemical bypass method, we deleted four of the FeS pathway proteins involved in sulfur acquisition and cluster assembly and demonstrated that they are all essential for parasite survival. However, the effect that these deletions had on the apicoplast organelle differed. Deletion of the cysteine desulfurase SufS led to disruption of the apicoplast organelle and loss of the organellar genome, whereas the other deletions did not affect organelle maintenance. Ultimately, we discovered that the requirement of SufS for organelle maintenance is not driven by its role in FeS biosynthesis, but rather, by its function in generating sulfur for use by MnmA, a tRNA modifying enzyme that we localized to the apicoplast. Complementation of MnmA and SufS activity with a bacterial MnmA and its cognate cysteine desulfurase strongly suggests that the parasite SufS provides sulfur for both FeS biosynthesis and tRNA modification in the apicoplast. The dual role of parasite SufS is likely to be found in other plastid-containing organisms and highlights the central role of this enzyme in plastid biology.

## Editor's evaluation

This study provides important new insights into iron sulfur biosynthesis in the human malaria parasite *Plasmodium falciparum*. The work is based on elegant and robust genetic approaches, and not only confirms the essentiality of the plastid-hosted Suf iron-sulfur cluster synthesis pathway, but also highlights an important additional role for the cysteine desulfurase SufS in apicoplast maintenance via tRNA modification. The work provides compelling evidence for a dual function of parasite SufS, although impact on tRNA has not been established directly. These findings reveal a potential new target for metabolic intervention and will be of interest to researchers studying apicomplexan parasites, and more broadly, in the field of plastid biology.

## Introduction

Malaria parasites contain a relict plastid organelle called the apicoplast that is required for its survival (*Köhler et al., 1997*; *McFadden et al., 1996*). The essentiality of this organelle and the unique

biochemical pathways within, such as iron-sulfur cluster (FeS), and isoprenoid precursor biosynthetic pathways offers a potentially rich source of new antimalarial drug targets (*Ellis et al., 2001*; *Jomaa et al., 1999*; *Seeber, 2002*). Since its discovery, several inhibitors targeting these and other pathways have been described, supporting the assertion that this organelle represents a viable source of novel drug targets (*Botté et al., 2012*; *Dahl and Rosenthal, 2008*; *Ke et al., 2014*; *Shears et al., 2015*).

FeS serve as cofactors for an array of proteins across kingdoms and are involved in a myriad of biological functions, including electron transfer, sulfur donation, redox sensing, gene expression, and translation (*Blahut et al., 2020*; *Przybyla-Toscano et al., 2018*; *Rouault, 2019*). FeS cofactors are found in a variety of forms, most commonly in the rhombic 2Fe-2S or the cubic 4Fe-4S forms and are typically bound to proteins through covalent bonds with cysteine side chains (*Beinert, 2000*; *Lill, 2009*; *Lu, 2018*). In *Plasmodium falciparum*, FeS cofactors are formed within the apicoplast by the sulfur utilization factor (SUF) pathway, for use by FeS-dependent proteins within the organelle (*Charan et al., 2017*; *Gisselberg et al., 2013*; *Pala et al., 2018*; *Swift et al., 2022*), while the mitochondrion houses the iron-sulfur cluster formation (ISC) pathway (*Dellibovi-Ragheb et al., 2013*; *Gisselberg et al., 2013*; *Sadik et al., 2021*). The ISC pathway generates FeS for use by FeS-dependent proteins within the mitochondrion, in addition to transferring a sulfur-containing moiety to the cytosolic iron-sulfur protein assembly machinery for FeS generation and transfer to cytosolic and nuclear proteins (*Dellibovi-Ragheb et al., 2013*; *Lill, 2009*).

FeS biosynthesis is organized into three steps: sulfur acquisition, cluster assembly, and cluster transfer (*Figure 1A*). The SUF pathway employs the cysteine desulfurase (SufS) to mobilize sulfur from L-cysteine, resulting in a SufS-bound persulfide (*Black and Dos Santos, 2015b*; *Loiseau et al., 2003*; *Ollagnier-de-Choudens et al., 2003*). SufE can enhance the cysteine desulfurase activity of SufS (*Nmu et al., 2007*; *Outten et al., 2003*; *Pilon-Smits et al., 2002*; *Wollers et al., 2010*; *Ye et al., 2006*) and is also able to accept sulfur atoms from SufS and transfer them to the SufBC$_2$D FeS assembly complex (*Saini et al., 2010*). The SufBC$_2$D complex serves as a scaffold for cluster formation with the ATPase activity of SufC being essential for the accumulation of iron from an unknown source (*Bai et al., 2018*; *Hirabayashi et al., 2015*; *Saini et al., 2010*). Subsequently, assembled clusters are transferred to downstream target apoproteins by transfer proteins such as SufA and NfuA (*Chahal et al., 2009*; *Py et al., 2012*).

In *Plasmodium*, all components of the SUF system are nuclear-encoded and trafficked to the apicoplast, except SufB, which is encoded by the ~35 kb apicoplast organellar genome (*Wilson et al., 1996*). The localization and activity of multiple components of the SUF pathway from *Plasmodium* spp. have been confirmed, including the cysteine desulfurase activity of SufS, its interaction with SufE, and their localization to the apicoplast (*Charan et al., 2014*; *Gisselberg et al., 2013*). In vitro studies demonstrated the ATPase activity of SufC and complex formation by SufB, SufC, and SufD (*Charan et al., 2017*; *Kumar et al., 2011*). Similarly, the cluster transfer proteins SufA and NfuApi (an ortholog of NufA) were shown to bind FeS cofactors and transfer them to a model acceptor protein (*Charan et al., 2017*). In the apicoplast, the acceptor proteins are presumably IspG and IspH (enzymes in the isoprenoid precursor biosynthesis pathway), ferredoxin (Fd), LipA (lipoic acid synthase), and MiaB (tRNA methylthiotransferase) (*Akuh et al., 2022*; *Gisselberg et al., 2013*; *Ralph et al., 2004*; *Swift et al., 2022*).

The SUF pathway appears to be required for the development of mosquito-stage parasites since conditional depletion of SufS blocks sporozoite maturation in *Plasmodium berghei* (*Charan et al., 2017*). The essentiality of SUF pathway proteins in blood-stage parasites is less clear since attempts to delete SufS, SufE, SufC, and SufD in *P. berghei* were not successful (*Haussig et al., 2014*). SufA and NfuApi are individually dispensable in blood-stage *P. falciparum* and *P. berghei* (*Haussig et al., 2013*; *Swift et al., 2022*), but display synthetic lethality when both are deleted (*Swift et al., 2022*). These results and the toxic effects of a dominant-negative SufC mutant (*Gisselberg et al., 2013*) suggest that the SUF pathway is essential for blood-stage parasites.

Blood-stage *P. falciparum* parasites can survive without an apicoplast as long as sufficient isopentenyl pyrophosphate (IPP), a product of the apicoplast isoprenoid biosynthetic pathway, is exogenously provided to the parasite (*Yeh and DeRisi, 2011*). Since this discovery, the IPP chemical bypass system has been used to investigate the role and essentiality of numerous apicoplast-specific proteins and pathways, including the SUF pathway (*Gisselberg et al., 2013*) and FeS-dependent proteins in the apicoplast (*Swift et al., 2022*). In the work outlined here, we determined the roles of SUF

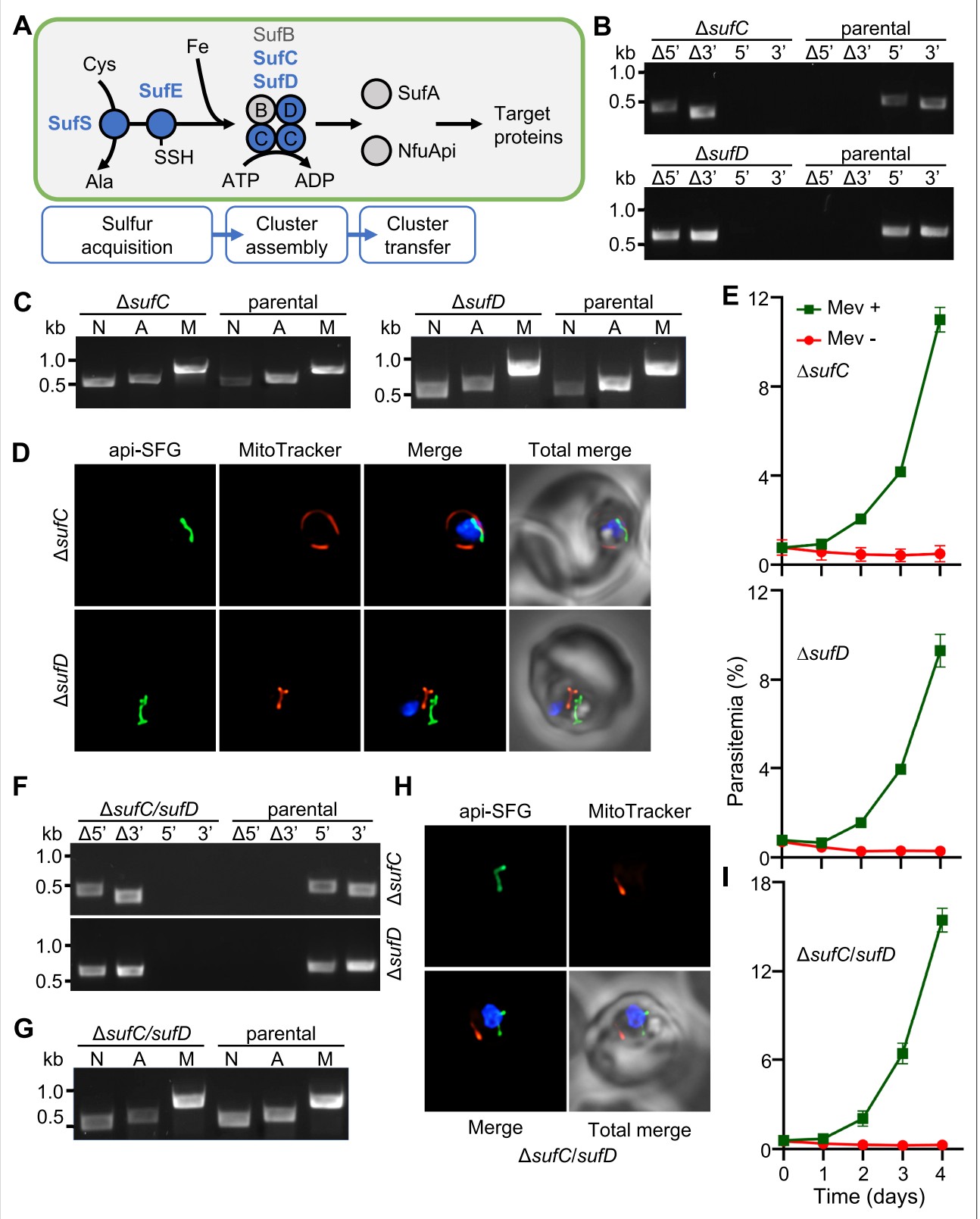

**Figure 1.** Apicoplast FeS assembly is not essential for apicoplast maintenance. (**A**) FeS assembly in the *P. falciparum* apicoplast. In the sulfur acquisition step, SufS liberates sulfur from cysteine (Cys), resulting in a bound persulfide. SufE transfers sulfur from SufS to SufB of the SufBC₂D FeS assembly complex. SufA and NfuApi receive FeS cofactors from the SufBC₂D complex and transfer them to downstream FeS-dependent proteins. The proteins investigated in this study are colored blue. Ala, alanine; Fe, iron. (**B**) Genotyping PCR confirming deletion of *sufC* (top panel) and Δ*sufD* (bottom

*Figure 1 continued on next page*

*Figure 1 continued*

panel). (**C**) Successful amplification of *sufB* (lane A) gene in the Δ*sufC* (left panel) and Δ*sufD* (right panel) parasites indicates the presence of the apicoplast genome. (**D**) Representative epifluorescence microscopy images of Δ*sufC* (top panel) and Δ*sufD* (bottom panel) parasites show a single intact apicoplast. (**E**) Mevalonate (Mev)-dependent growth of Δ*sufC* (top panel) and Δ*sufD* (bottom panel) parasites over 4 days. (**F**) Genotyping PCR confirms deletion of both *sufC* (top panel) and *sufD* (bottom panel) in Δ*sufC/sufD* double knockout parasites. (**G**) Successful amplification of *sufB* (lane A) demonstrates intact apicoplast in Δ*sufC/sufD* parasites. (**H**) Representative epifluorescence microscopy images of Δ*sufC/sufD* parasites shows an apicoplast with typical intact morphology. (**I**) Mevalonate (Mev)-dependent growth of Δ*sufC/sufD* parasites over 4 days. In panels (**B**) and (**F**), successful gene deletion is demonstrated by the presence of PCR products for the Δ5′ and Δ3′ loci, but not the unmodified loci (5′ and 3′) found in the PfMev (parental) line. Expected PCR amplicon sizes are in *Figure 1—figure supplement 1C*. In (**C**) and (**G**), PCR detection of the *ldh*, *sufB*, and *cox1* genes of the parasite nuclear (N), apicoplast (A), and mitochondrial (M) genomes, respectively, were attempted from transgenic as well as PfMev (parental) parasite lines. In (**D**) and (**H**), the api-SFG protein (green) labels apicoplast, MitoTracker (red) stains the mitochondrion, and nuclear DNA is stained with DAPI (blue). Each image shows a field representing 10 μm × 10 μm. For panels (**E**) and (**I**), asynchronous parasites were cultured with or without 50 μM Mev and parasitemia was monitored every 24 hr via flow cytometry. Data points show daily mean parasitemia ± standard error of the mean (SEM) from 2 independent biological replicates, each with four technical replicates. DNA markers in (**B**), (**C**), (**F**), and (**G**) are in kilobases (kb).

The online version of this article includes the following source data and figure supplement(s) for figure 1:

**Source data 1.** Uncropped agarose gel images of PCR analyses presented in *Figure 1B, C, F, and G*.

**Source data 2.** Growth assay parasitemia counts for Δ*sufC* (top table) and Δ*sufD* (bottom table) used for *Figure 1E*.

**Source data 3.** Δ*sufC/sufD* growth assay parasitemia counts used for *Figure 1(I)*.

**Figure supplement 1.** Generation and characterization of knockout parasite lines.

**Figure supplement 1—source data 1.** Table used for *Figure 1—figure supplement 1B*.

**Figure supplement 1—source data 2.** Table used for *Figure 1—figure supplement 1C*.

pathway proteins in an apicoplast metabolic bypass parasite line containing a genetically encoded isoprenoid precursor biosynthesis bypass pathway (*Swift et al., 2020*). We found that SUF proteins involved in sulfur acquisition (SufS and SufE) and cluster assembly (SufC and SufD) are all essential for parasite survival. Deletion of the cysteine desulfurase SufS, however, resulted in loss of the apicoplast organelle and its organellar genome whereas deletion of the other SUF proteins did not result in this phenotype. We hypothesized that SufS is also responsible for providing sulfur for use by MnmA, a tRNA modifying enzyme. We found that MnmA is located in the apicoplast and its loss results in the same apicoplast disruption phenotype as observed for SufS. Complementation experiments using *Bacillus subtilis* MnmA and its cognate cysteine desulfurase strongly suggest that SufS is required for both FeS synthesis and tRNA modification by MnmA, a novel paradigm that is likely to apply to other plastid-containing organisms.

## Results

### FeS assembly complex proteins SufC and SufD are essential for parasite survival, but are not required for apicoplast maintenance

In *P. falciparum*, FeS assembly in the apicoplast is mediated by the SufBC$_2$D complex (*Charan et al., 2017*). SufB (PF3D7_API04700) is encoded by the apicoplast genome, while SufC (PF3D7_1413500) and SufD (PF3D7_1103400) are encoded by the nuclear genome. Dominant negative experiments with SufC suggested that the complex is essential for parasite survival (*Gisselberg et al., 2013*), however, gene deletions have not been successful in any malaria parasite species. To investigate the role of the FeS assembly complex, we targeted *sufC* and *sufD* in a metabolic bypass parasite line (PfMev) under continuous mevalonate supplementation (*Figure 1—figure supplement 1*). PfMev parasites synthesize IPP from exogenously provided mevalonate, enabling the disruption of genes encoding essential apicoplast-specific proteins (*Swift et al., 2020*). We confirmed the successful deletion of both *sufC* and *sufD* genes using genotyping PCR (*Figure 1B*). To determine whether the deletion of *sufC* or *sufD* resulted in apicoplast disruption, we attempted to amplify the apicoplast genome-encoded SufB from both the Δ*sufC* and Δ*sufD* parasite lines. Both parasite lines contained the *sufB* gene, indicating retention of the apicoplast genome (*Figure 1C*). We also observed intact apicoplast organelles, labeled by an apicoplast-localized super-folder green protein reporter (api-SFG) (*Swift et al., 2020*), in these parasite lines by live epifluorescence microscopy (*Figure 1D*). Despite the presence of intact apicoplast organelles, both parasite lines required supplementation with exogenous

mevalonate, demonstrating that SufC and SufD are essential proteins (*Figure 1E*). Taken together, these results show that SufC and SufD are essential for parasite survival, although neither protein is required for apicoplast maintenance.

Since SufC and SufD are essential, it is likely that the $SufBC_2D$ complex is essential. Unfortunately, we cannot genetically modify the apicoplast genome-encoded SufB with any available experimental techniques, but we can simultaneously delete both SufC and SufD to remove the possibility of any type of complex forming with SufB. In *Escherichia coli*, pulldown assays demonstrated that different types of FeS assembly complexes can be formed ($SufBC_2D$ and $SufB_2C_2$), which suggests that there may be some redundancy between complex proteins (*Saini et al., 2010*; *Yuda et al., 2017*). We generated Δ*sufC/sufD* double knockout parasites in the PfMev line, under continuous supplementation of mevalonate (*Figure 1F*). Consistent with the phenotypes of the Δ*sufC* and Δ*sufD* parasite lines, Δ*sufC/sufD* parasites were also found to have intact apicoplasts (*Figure 1G, H*) and were dependent on exogenous mevalonate supplementation for survival (*Figure 1I*). Collectively, these findings demonstrate that while the FeS assembly complex is essential for parasite survival, it is not required for apicoplast maintenance.

## SufS is required for apicoplast maintenance while SufE is not

We next investigated the sulfur acquisition steps of the SUF pathway upstream of SufC and SufD. In the apicoplast of *P. falciparum*, SufS (PF3D7_0716600), along with its partner SufE (PF3D7_0206100), mobilize sulfur from L-cysteine, with SufE transferring the sulfur to the $SufBC_2D$ complex (*Gisselberg et al., 2013*). The cysteine desulfurase activity of the *P. falciparum* SufS has been confirmed biochemically (*Charan et al., 2014*) and through complementation in *E. coli* (*Gisselberg et al., 2013*). It was also shown that the *P. falciparum* SufE enhances the cysteine desulfurase activity of SufS by up to ~17-fold in an in vitro biochemical assay (*Charan et al., 2014*). To disrupt sulfur acquisition in the SUF pathway, we generated individual deletions of *sufS* and *sufE* in the PfMev line (*Figure 2A*). In Δ*sufE* parasites, the apicoplast remained intact as evidenced by successful PCR amplification of the *sufB* gene; however, we were unable to amplify this gene from Δ*sufS* parasites (*Figure 2B*). Consistent with the PCR results, we observed an intact apicoplast in Δ*sufE* parasites by live microscopy, while in Δ*sufS* parasites we observed multiple discrete api-SFG labeled vesicles - a hallmark of apicoplast organelle disruption (*Figure 2C*). Additionally, both parasite lines were dependent on mevalonate for survival (*Figure 2D*). Taken together, these results indicate that both SufE and SufS are required for parasite survival, but only SufS is required for apicoplast maintenance.

## MnmA is essential for apicoplast maintenance

While SufS is required for apicoplast maintenance, none of the other SUF pathway proteins or any of the FeS-dependent proteins in the apicoplast are required for this process (*Swift et al., 2022*). This suggests that the reliance on sulfur for organelle maintenance is likely driven by a different sulfur-dependent pathway. Several biochemical pathways require sulfur, including those involved in the biosynthesis of thiamine, biotin, lipoic acid, molybdopterin, and thio-modifications of tRNA (*Hidese et al., 2011*; *Leimkühler et al., 2017*; *Mihara and Esaki, 2002*). Of these pathways, we found that the biosynthesis of lipoic acid and tRNA thio-modifications were the only ones that appeared to be present in *P. falciparum* with predicted localization to the apicoplast (*Ralph et al., 2004*; *Supplementary file 1*a). In a recent study, we showed that lipoic acid synthesis is dispensable (*Swift et al., 2022*), however, tRNA thiolation has not been studied in malaria parasites. Based on sequence homology, *P. falciparum* parasites appear to contain several enzymes capable of catalyzing tRNA thiolation reactions, but only one appears to be a possible apicoplast protein. This protein (PF3D7_1019800) is currently annotated as a tRNA methyltransferase (*Aurrecoechea et al., 2008*), but it shares 25% sequence identity with an *E. coli* enzyme called MnmA (tRNA-specific 2-thiouridylase) (*Figure 3—figure supplement 1*). MnmA inserts sulfur at carbon-2 (C2) of uridine at position 34 (s2U34) of tRNA$^{Lys}_{UUU}$, tRNA$^{Glu}_{UUC}$, and tRNA$^{Gln}_{UUG}$ (*Black and Dos Santos, 2015a*; *Leimkühler et al., 2017*; *Shigi, 2014*; *Shigi, 2018*). *E. coli* MnmA receives sulfur from a series of five sulfur transfer proteins, TusA/B/C/D/E (*Black and Dos Santos, 2015a*; *Ikeuchi et al., 2006*; *Shigi, 2014*), which ultimately acquire sulfur from the IscS cysteine desulfurase, but cannot obtain sulfur from the *E. coli* SufS (*Buhning et al., 2017*). We could not identify any orthologs of the Tus proteins by homology searches in *P. falciparum*, and

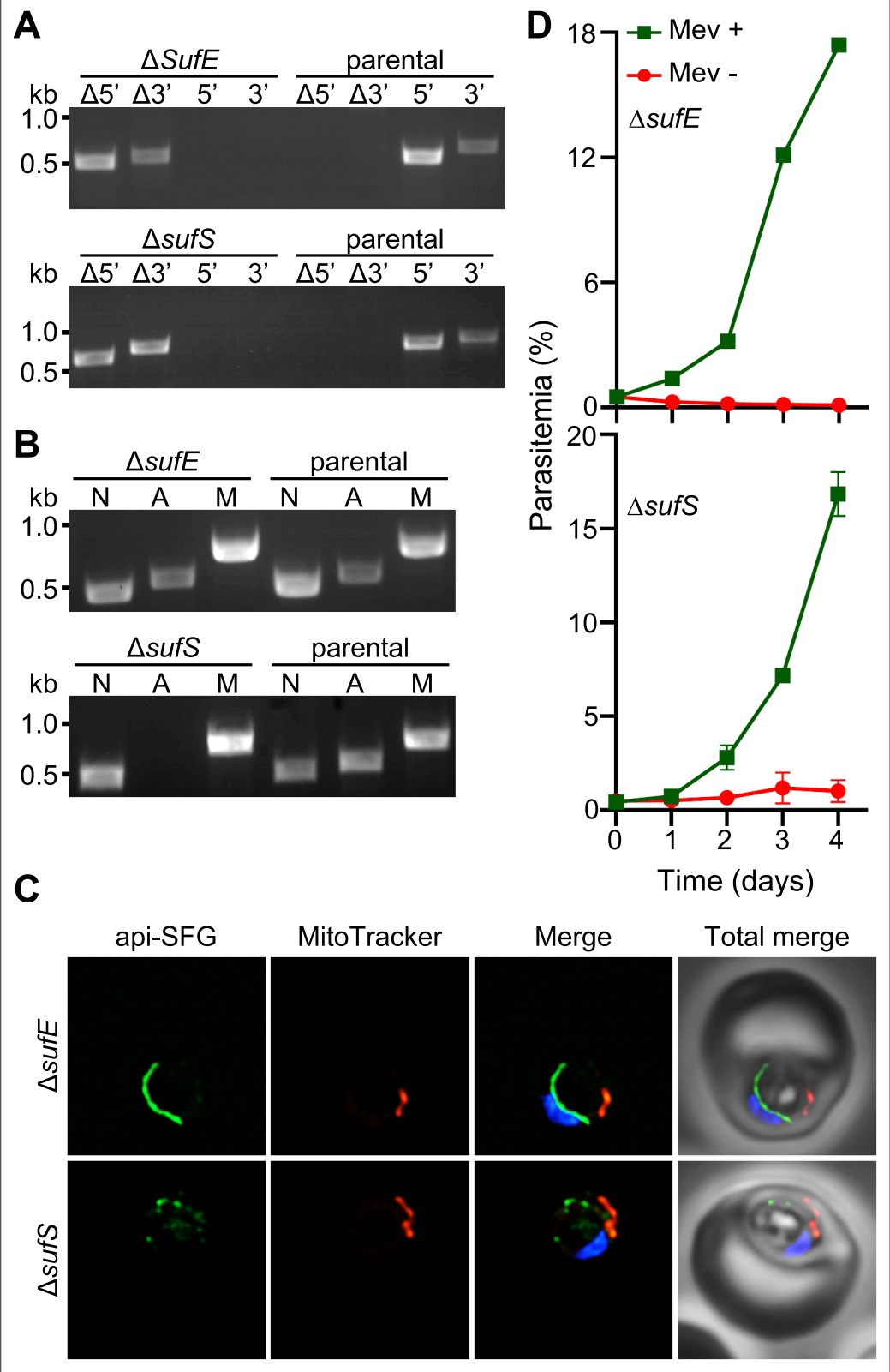

**Figure 2.** Apicoplast cysteine desulfurase, SufS, is essential for apicoplast maintenance. (**A**) Genotyping PCR confirming deletion of *sufE* (top panel) or *sufS* (bottom panel) in PfMev parasites. In both Δ*sufE* and Δ*sufS* parasite lines, gene deletions were validated by the presence of PCR products for the Δ5' and Δ3' loci, but not for the unmodified loci (5' and 3') found in the PfMev (parental) line. Genotyping PCRs and expected amplicon sizes are

*Figure 2 continued on next page*

*Figure 2 continued*

described in *Figure 1—figure supplement 1*. (**B**) Attempted PCR amplification of *ldh*, *sufB*, and *cox1* genes of the parasite nuclear (**N**), apicoplast (**A**), and mitochondrial (**M**) genomes, respectively, from the Δ*sufE*, Δ*sufS*, and PfMev (parental) parasites. Successful amplification of *sufB* in the Δ*sufE* (top panel) parasites indicates the presence of the apicoplast genome, while failure to amplify this gene in Δ*sufS* parasites (bottom panel) indicates loss of the apicoplast genome. (**C**) Representative epifluorescence microscopy images of Δ*sufE* (top panel) and Δ*sufS* (bottom panel) parasites. In the Δ*sufE* parasites, a single intact apicoplast is seen. Whereas in Δ*sufS* multiple discrete vesicles were seen, indicating disruption of the apicoplast organelle. The api-SFG protein (green) labels the apicoplast, MitoTracker (red) stains the mitochondrion, and nuclear DNA is stained with DAPI (blue). Each image is a field representing 10 μm × 10 μm. (**D**) Mevalonate (Mev)-dependent growth of Δ*sufE* (top panel) and Δ*sufS* (bottom panel) parasites. Asynchronous parasites from each line were cultured with or without 50 μM Mev and parasitemia was monitored every 24 hr by flow cytometry for 4 days. Data points showing daily mean parasitemia ± SEM from two independent biological replicates, each with four technical replicates. In panels (**A**) and (**B**), DNA markers are in kilobases (kb).

The online version of this article includes the following source data for figure 2:

**Source data 1.** Uncropped agarose gel images of PCR analyses presented in *Figure 2A and B*.

**Source data 2.** Δ*sufE* (top table) and Δ*sufS* (bottom table) growth assay parasitemia counts used for *Figure 2D*.

the parasite IscS has already been localized to the mitochondrion instead of the apicoplast in malaria parasites (*Gisselberg et al., 2013*).

Multiple sequence alignments (MSA) of the putative *P. falciparum* MnmA (*Pf* MnmA) with orthologs from *Synechococcus* sp., *B. subtilis*, *E. coli*, *Arabidopsis thaliana*, and *Saccharomyces cerevisiae* reveal that it has a 421 amino acid (aa) N-terminal extension (*Figure 3—figure supplement 1*), which is predicted to contain an apicoplast transit peptide (*Foth et al., 2003*; *Ralph et al., 2004*). MSA also demonstrates that the *Pf* MnmA has conserved catalytic cysteines as well as a highly conserved ATP-binding PP-loop motif (SGGXDS) (*Figure 3A*, *Figure 3—figure supplement 1*; *Numata et al., 2006*; *Shigi et al., 2020*). This PP-loop motif activates the C2 of nucleotide U34 of the target tRNA by adenylation in an ATP-dependent manner (*Mueller, 2006*; *Numata et al., 2006*). The first catalytic cysteine receives sulfur generating an MnmA-persulfide, while the second catalytic cysteine releases the sulfur from the adduct and transfers it to the activated U34 (*Čavužić and Liu, 2017*; *Shigi et al., 2020*). Other pathogenic apicomplexans also seem to have an ortholog of MnmA with the conserved cysteines and domains (*Figure 3—figure supplement 2*) and the ortholog from *Toxoplasma gondii* (ToxoDB ID: TGME49_309110) has recently been described (*Yang et al., 2022*). In silico structural analysis of the C-terminal region of *Pf* MnmA (residues 422–1084) and the corresponding regions of other apicomplexan MnmA orthologs predicts a MnmA-like fold (*Figure 3—figure supplement 3*).

To validate the predicted apicoplast localization of the putative *Pf* MnmA, we generated a parasite line with two C-terminal FLAG tags in tandem appended to the endogenous *Pf* MnmA (*Figure 3B, C and D*, *Figure 3—figure supplement 4*). This parasite line, *mnmA*-flag, also contains an aptamer array in the 3' untranslated region (UTR) of *mnmA* to use with the TetR-DOZI system (*Ganesan et al., 2016*; *Rajaram et al., 2020*) for inducible control over protein production (*Figure 3B*). We showed that FLAG-tagged *Pf* MnmA colocalizes with the apicoplast marker protein, acyl carrier protein (ACP) by immunofluorescence (Manders' coefficient, M1=0.818, standard deviation = ±0.187, n=22), confirming *Pf* MnmA localization to the apicoplast (*Figure 3E*). We next attempted to knockdown *Pf* MnmA using the TetR-DOZI system in the *mnmA*-flag parasite line (*Figure 3—figure supplement 5*). We monitored parasite growth in control (aTc added) and knockdown (aTc removed) conditions over 8 days. From day 5 onward, the parasites showed a significant growth defect under the knockdown condition (*Figure 3F*). When parasites in the knockdown condition were supplemented with mevalonate (rescue), the parasites grew similarly to parasites under the control condition (*Figure 3F*), further confirming the apicoplast-associated activity of *Pf* MnmA. During the growth assay, we also assessed the apicoplast morphology of parasites under each of the experimental conditions listed above via live epifluorescence microscopy every 48 hr. We started to observe multiple discrete api-SFG labeled vesicles at day 4 following aTc removal (*Figure 3G*, *Figure 3—figure supplement 6*), indicative of apicoplast disruption. At this point, about ~70% of parasites contained an intact apicoplast. By day 8, only ~25% of parasites had an intact apicoplast (*Figure 3H*, *Figure 3—figure supplement 7*). Significant growth defects and disruption of the apicoplast following knockdown of *Pf* MnmA

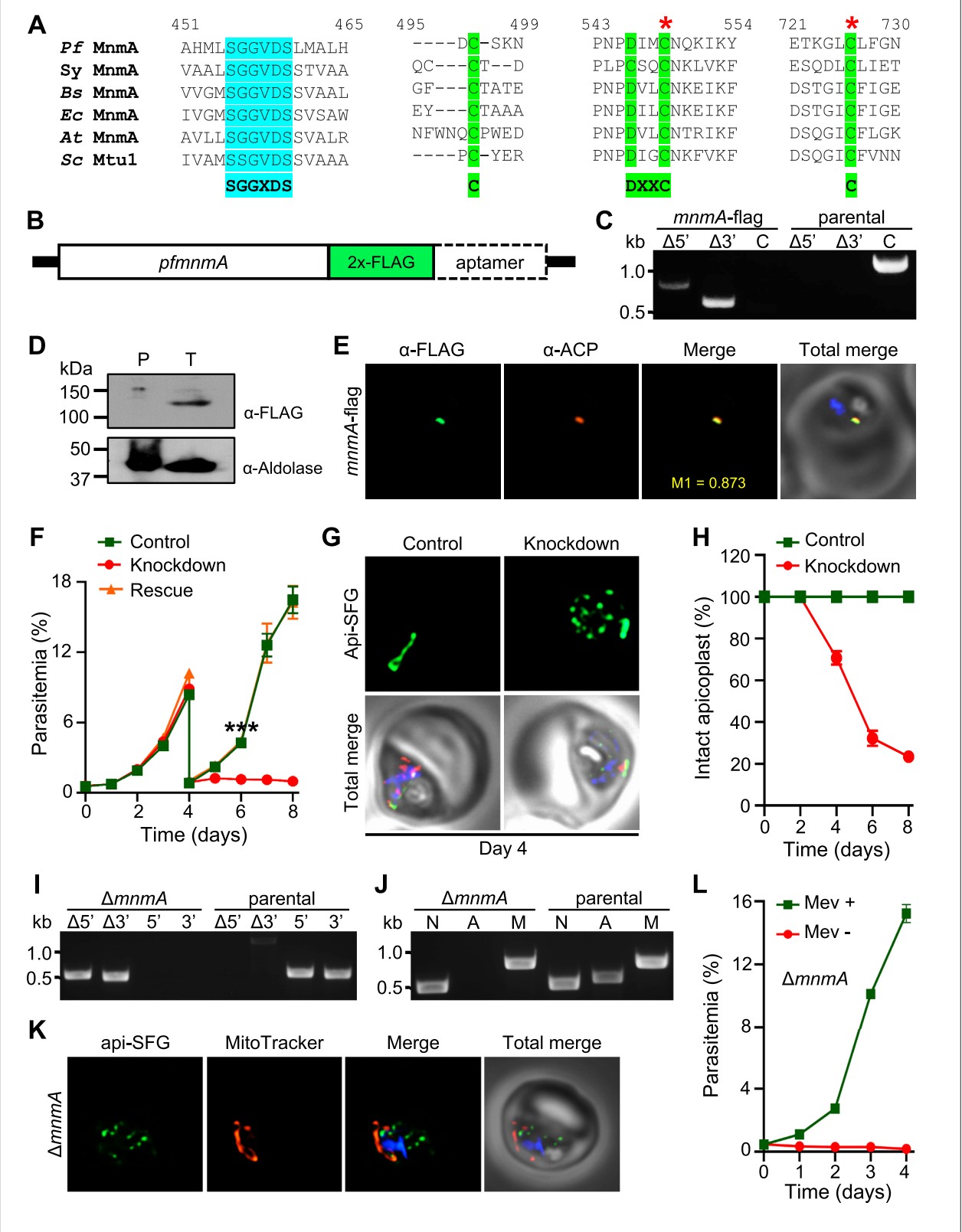

**Figure 3.** *Plasmodium falciparum* MnmA is essential for apicoplast maintenance and parasite survival. (**A**) Snapshots from the multiple sequence alignment of MnmA orthologs from *P. falciparum* (*Pf* MnmA), *Synechococcus* sp. (*Sy* MnmA), *B. subtilis* (*Bs* MnmA), *E. coli* (*Ec* MnmA), *A. thaliana* (*At* MnmA), and *S. cerevisiae* (*Sc* Mtu1) showing the conserved sequence motifs in the catalytic domains. The PP-loop motif SGGXDS shown in teal and the D-type motif C-DXXC-C in bright green. The red asterisks indicate catalytic cysteines. (**B**) Expected modified locus following introduction of 2xFLAG at

*Figure 3 continued on next page*

*Figure 3 continued*

3'-end of *pfmnmA* and 10xAptamer in the 3'UTR. (**C**) Genotyping of *mnmA*-flag parasites confirms plasmid integration as evidenced by the presence of PCR amplicons for the recombinant Δ5' and Δ3' loci but not for the unmodified locus (**C**), as compared to the PfMev^attB (parental) line. Expected PCR amplicon sizes are provided in *Figure 3—figure supplement 4B*. (**D**) Immunoblot of saponin-isolated PfMev^attB (parental, **P**) and *mnmA*-flag (tagged, **T**) parasite lysates with anti-FLAG antibody (top panel). Anticipated molecular weight for the tagged protein is 120 kDa. Anti-aldolase immunoblot shows relative loading levels (bottom panel). Protein markers are in kilodaltons (kDa). The uncropped version is shown in *Figure 3—figure supplement 4C*. (**E**) Representative immunofluorescence microscopy images show *Pf* MnmA-2xFLAG (green) colocalization with the apicoplast marker ACP (acyl carrier protein, red). Manders' coefficient (M1, green in red) quantifies the degree of colocalization. (**F**) Anhydrous tetracycline (aTc)-dependent growth of *mnmA*-flag parasites. Asynchronous parasites were grown with 0.5 μM aTc (control), without 0.5 μM aTc (knockdown), or with 50 μM mevalonate (Mev, rescue). Parasitemia was monitored via flow cytometry every 24 hr for 8 days. On day 4, parasite cultures were diluted 1:10. Data points show daily mean parasitemia ± SEM from two independent biological replicates, each with four technical replicates; two-way ANOVA (Sidak-Bonferroni method), ***p<0.001. (**G**) Representative epifluorescence microscopy images of day four samples of *mnmA*-flag parasites grown in control (aTc added) and knockdown (aTc removed) conditions from the experiment shown in (**F**). Under control conditions, the apicoplast remains intact; under knockdown conditions, multiple discrete vesicles are seen, demonstrating a disrupted apicoplast. The full image panel is shown in *Figure 3—figure supplement 6A*. Representative images for *mnmA*-flag parasites grown in the rescue condition (aTc removed, Mev added) from the experiment shown in (**F**) are presented in *Figure 3—figure supplement 6B*. (**H**) Graph showing the percentage of *mnmA*-flag parasites with an intact apicoplast when grown in control (aTc added) and knockdown (aTc removed) conditions in the experiment shown in (**F**). Percentage of intact apicoplast in the rescue condition (aTc removed, Mev added) also mirrors the trend observed in the knockdown condition. Data shown in *Figure 3—figure supplement 7*. Live epifluorescence microscopy images were taken every 48 hr and apicoplast morphology (intact or disrupted) was determined based on apicoplast-localized api-SFG. Error bars represent standard deviation from at least 18 images taken for each time point and growth condition from two independent experiments. (**I**) Genotyping PCR confirming *mnmA* deletion in Δ*mnmA* parasites. In the Δ*mnmA* parasite line, gene deletion was validated by the presence of PCR products for the Δ5' and Δ3' loci, but not for the unmodified loci (5' and 3') found in the PfMev (parental) line. Genotyping PCRs and expected amplicon sizes are described in *Figure 1—figure supplement 1*. (**J**) Attempted PCR amplification of *ldh*, *sufB*, and *cox1* genes of the parasite nuclear (N), apicoplast (A), and mitochondrial (M) genomes, respectively, from the Δ*mnmA* and PfMev (parental) parasites. Lack of a PCR amplicon for *sufB* in the Δ*mnmA* parasite line suggests loss of the apicoplast genome. (**K**) Representative epifluorescence microscopy images of Δ*mnmA* parasites. Multiple discrete vesicles are seen, demonstrating a disrupted apicoplast. (**L**) Mevalonate (Mev)-dependent growth of the Δ*mnmA* parasites. Asynchronous parasites were cultured with or without 50 μM Mev for 4 days. Parasitemia was monitored every 24 hr by flow cytometry. Data points show daily mean parasitemia ± SEM from two independent biological replicates, each with four technical replicates. In panels (**G**) and (**K**), api-SFG protein (green) labels the apicoplast, the mitochondrion is stained with MitoTracker (red), and nuclear DNA is stained with DAPI (blue). In (**E**), (**G**), and (**K**), each image depicts a field of 10 μm × 10 μm. In (**C**), (**I**), and (**J**), DNA markers are in kilobases (kb).

The online version of this article includes the following source data and figure supplement(s) for figure 3:

**Source data 1.** Uncropped agarose gel images of PCR analyses presented in *Figure 3C, I, and J*.

**Source data 2.** *mnmA*-flag growth assay parasitemia counts used for *Figure 3F*.

**Source data 3.** Counts of intact apicoplasts presented in *Figure 3(H)*.

**Source data 4.** Δ*mnmA* growth assay parasitemia counts used for *Figure 3L*.

**Figure supplement 1.** Sequence identity among MnmA orthologs.

**Figure supplement 1—source data 1.** Multiple sequence alignment used for *Figure 3A* and *Figure 3—figure supplement 1*.

**Figure supplement 1—source data 2.** Table used for *Figure 3—figure supplement 1A*.

**Figure supplement 2.** MnmA orthologs from pathogenic apicomplexans.

**Figure supplement 2—source data 1.** Multiple sequence alignment used for *Figure 3—figure supplement 2*.

**Figure supplement 2—source data 2.** Table used for *Figure 3—figure supplement 2B*.

**Figure supplement 3.** The C-terminal domain of *P. falciparum* MnmA is predicted to have the same protein fold as bacterial MnmA.

**Figure supplement 3—source data 1.** Table used for *Figure 3—figure supplement 3D*.

**Figure supplement 3—source data 2.** Multiple sequence alignment used for *Figure 3—figure supplement 3E*.

**Figure supplement 4.** Generation of the MnmA localization/knockdown construct.

**Figure supplement 4—source data 1.** Table used for *Figure 3—figure supplement 4B*.

**Figure supplement 4—source data 2.** Full immunoblot shown in *Figure 3—figure supplement 4C*.

**Figure supplement 5.** Effect of aTc removal on *Pf* MnmA in the *mnmA*-flag line.

**Figure supplement 5—source data 1.** Full immunoblot shown in *Figure 3—figure supplement 5*.

**Figure supplement 6.** Live epifluorescence microscopy shows the effect on apicoplast morphology in the *mnmA*-flag line after aTc removal.

**Figure supplement 7.** Percentage of intact apicoplast in the *mnmA*-flag line after aTc removal.

**Figure supplement 7—source data 1.** Counts of intact apicoplasts presented in *Figure 3—figure supplement 7*.

**Figure supplement 8.** The N-terminal region of *Pf* MnmA contains a predicted SufE-like domain.

**Figure supplement 8—source data 1.** Table used for *Figure 3—figure supplement 8F*.

**Figure supplement 8—source data 2.** Table used for *Figure 3—figure supplement 8H*.

*Figure 3 continued*

**Figure supplement 8—source data 3.** Multiple sequence alignment used for *Figure 3—figure supplement 8H*.

suggest that *Pf* MnmA is essential for parasite survival and apicoplast maintenance. To further confirm these findings, we deleted the *mnmA* gene through Cas9-mediated genome editing in PfMev parasites under continuous mevalonate supplementation (*Figure 3I*). The deletion of *mnmA* resulted in apicoplast disruption, as evidenced by the inability to detect the apicoplast genome encoded *sufB* gene (*Figure 3J*) and the presence of multiple discrete vesicles labeled by api-SFG (*Figure 3K*). Additionally, Δ*mnmA* parasites were dependent on exogenous mevalonate supplementation for survival (*Figure 3L*), which can be explained by the loss of *sufB* resulting in defective FeS synthesis and the inability to synthesize isoprenoids (*Akuh et al., 2022*; *Elahi and Prigge, 2023*; *Swift et al., 2022*). Overall, these results demonstrate that *Pf* MnmA is required for both apicoplast maintenance and parasite survival.

### *Bacillus subtilis* MnmA and YrvO can be expressed in the *P. falciparum* apicoplast

In *E. coli*, $s^2$U biosynthesis starts with the acquisition of sulfur from L-cysteine by the cysteine desulfurase IscS, which then relays the sulfur via the five proteins of the Tus system (TusABCDE) to MnmA (*Black and Dos Santos, 2015a*; *Ikeuchi et al., 2006*; *Outten et al., 2003*; *Shigi, 2014*). MnmA then uses that sulfur to modify the target tRNAs at the U34 position in an ATP-dependent manner (*Mueller, 2006*; *Numata et al., 2006*). Not all bacteria contain IscS or the Tus system to relay sulfur to MnmA. In *B. subtilis* for example, a specialized cysteine desulfurase, YrvO, provides sulfur directly to MnmA (*Black and Dos Santos, 2015a*). Although there are four different cysteine desulfurases (all four are SufS paralogs) present in *B. subtilis*, both in vivo complementation and in vitro biochemical studies established that YrvO is the sulfur source for the $s^2$U modification catalyzed by MnmA (*Black and Dos Santos, 2015a*). To clarify the role of MnmA in *P. falciparum,* we attempted to complement *Pf* MnmA with the *B. subtilis* (*Bs*) MnmA in PfMev^attB parasites (*Swift et al., 2021*), through knock-in via myco-bacteriophage integrase-mediated recombination (*Figure 4—figure supplement 1*; *Spalding et al., 2010*). This parasite line is hereafter referred to as *bsmnmA*^+. *Bs* MnmA might not be functional in *P. falciparum* in the absence of its cognate cysteine desulfurase, *Bs* YrvO, as previously demonstrated in *E. coli* (*Black and Dos Santos, 2015a*). To address this possibility, we also generated a parasite line expressing a *Bs* MnmA-YrvO fusion protein (*bsmnmA-yrvO*^+) (*Figure 4—figure supplement 2*) using the same knock-in method.

The *bsmnmA* and *bsmnmA-yrvO* expression cassettes encode a conditional localization domain (CLD) at the protein N-terminus for inducible control over protein localization (*Roberts et al., 2019*) and contain an mCherry tag on the C-terminus for visualization by live cell fluorescence (*Figure 4A*). The CLD directs the tagged protein to the apicoplast, but following the addition of the ligand, *Shield1*, the tagged protein is redirected to the parasitophorous vacuole (*Roberts et al., 2019*). An aptamer array was also included at the 3' UTR of these genes, for use with the TetR-DOZI system (*Ganesan et al., 2016*; *Rajaram et al., 2020*; *Figure 4A*). We successfully generated both knock-in lines (*Figure 4B*). Live epifluorescence microscopy for both lines showed that *Bs* MnmA and *Bs* MnmA-YrvO proteins are trafficked to the apicoplast, as evidenced by the colocalization of the mCherry signal with the apicoplast api-SFG signal (*Figure 4C*; Manders' coefficient, M1 of *bsmnmA*^+=0.757, standard deviation = ± 0.155, n=14; M1 of *bsmnmA-yrvO*^+=0.740, standard deviation = ± 0.142, n=15). Taken together, these results show that *Bs* MnmA and *Bs* MnmA-YrvO can be expressed in the apicoplast of *P. falciparum.*

### *Bs* MnmA-YrvO fusion protein can complement the loss of parasite MnmA

We used *bsmnmA-yrvO*^+ parasites to investigate whether loss of *Pf* MnmA can be complemented by the *Bs* MnmA-YrvO fusion protein. We employed the same gRNA and repair construct used in the experiment described in *Figure 3* to delete *pfmnmA* in the *bsmnmA-yrvO*^+ line. In repeated independent transfection experiments, we successfully generated *bsmnmA-yrvO*^+ Δ*mnmA* parasites (*Figure 5A*, *Supplementary file 1*b). These parasites were mevalonate independent (*Figure 5B*) and

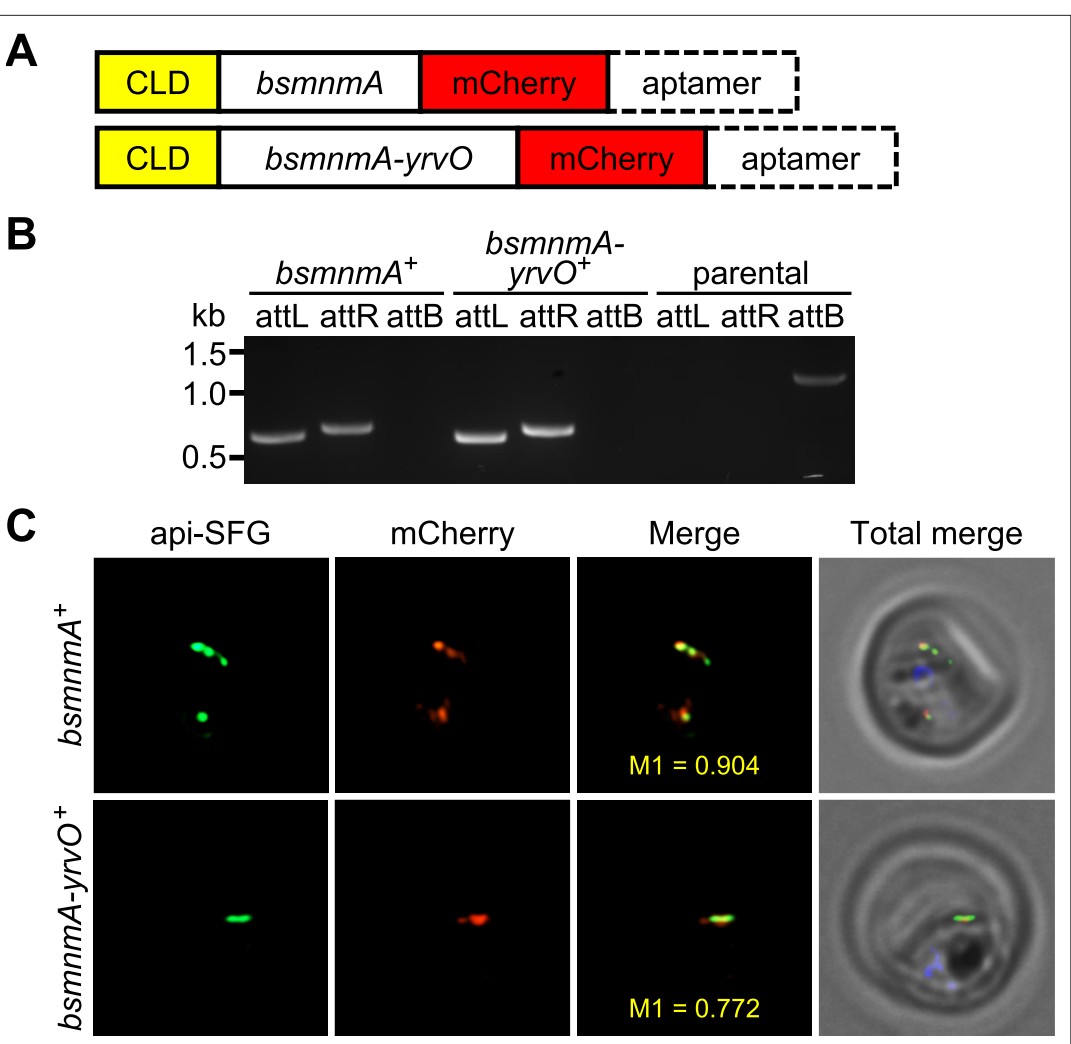

**Figure 4.** *Bacillus subtilis* MnmA and MnmA-YrvO can be expressed in the parasite apicoplast. (**A**) Schematics illustrating the expected gene product in the *bsmnmA⁺* and *bsmnmA-yrvO⁺* parasite lines following successful genetic modification. CLD, conditional localization domain. (**B**) Amplification of the attL and attR regions from *bsmnmA⁺* and *bsmnmA-yrvO⁺* transgenic parasites indicates successful plasmid integration in both lines. Amplification of the attB region from the PfMev^attB parasite line (parental) was used as a control. DNA markers are in kilobases (kb). Expected PCR amplicon sizes are provided in *Figure 4—figure supplement 1B*. (**C**) Representative epifluorescence microscopy images of *bsmnmA⁺* (top panel) and *bsmnmA-yrvO⁺* (bottom panel) transgenic parasites. The *bsmnmA* and *bsmnmA-yrvO* fusion protein contains a C-terminal mCherry tag (red) and colocalizes with the apicoplast api-SFG marker (green). The nuclear DNA is stained with DAPI (blue). The degree of colocalization of the *bsmnmA* or *bsmnmA-yrvO* fusion proteins with the api-SFG apicoplast marker are shown in the merge images and the Manders' coefficients (M1; red in green) are provided. Each image depicts a field of 10 μm ×10 μm.

The online version of this article includes the following source data and figure supplement(s) for figure 4:

**Source data 1.** Uncropped agarose gel images of PCR analyses presented in *Figure 4B*.

**Figure supplement 1.** Generation and characterization of parasite lines expressing *B. subtilis* MnmA and MnmA-YrvO.

**Figure supplement 1—source data 1.** Table used for *Figure 4—figure supplement 1B*.

**Figure supplement 2.** The engineered sequence encoding the *B. subtilis* MnmA-YrvO fusion protein.

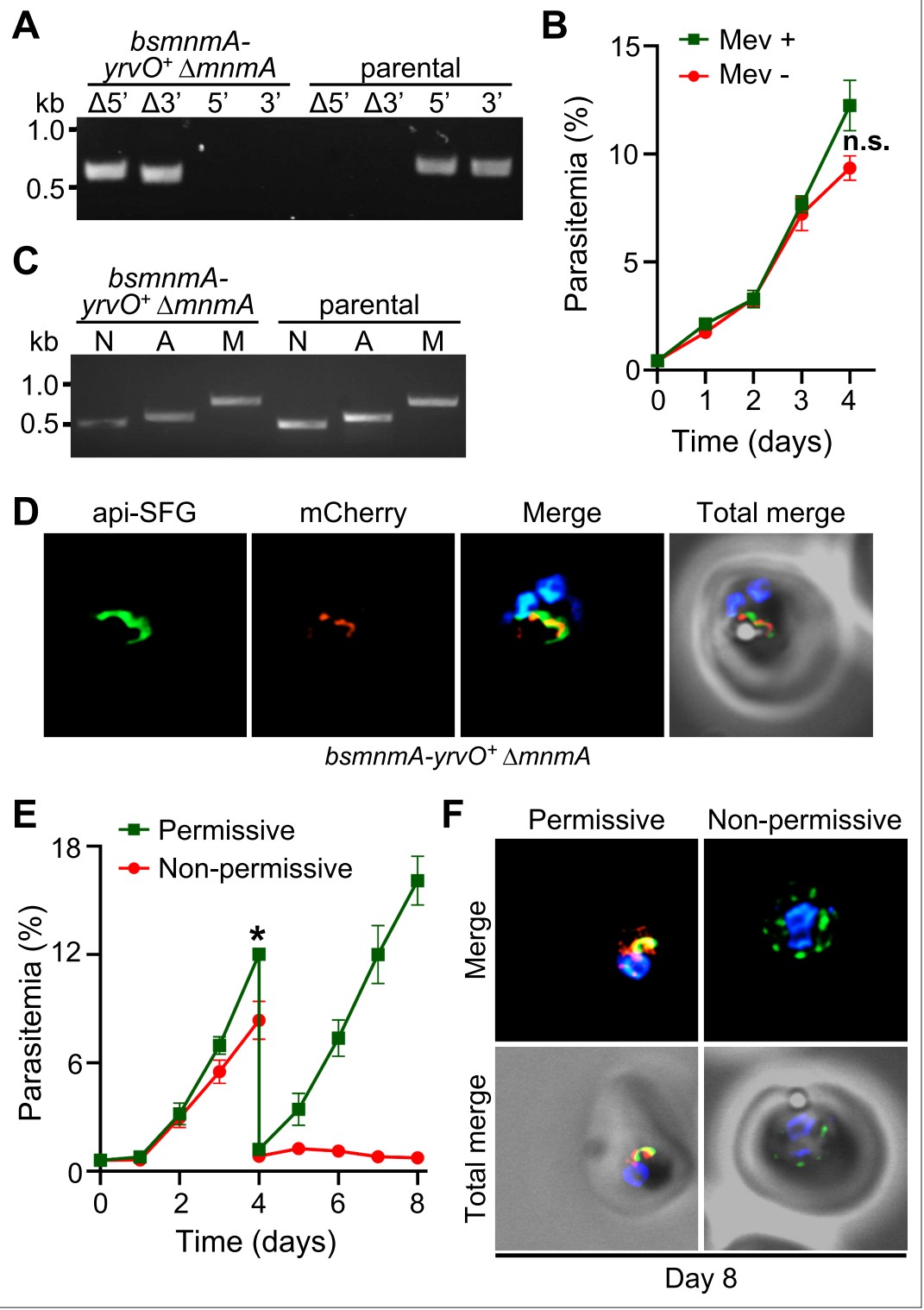

**Figure 5.** *Bacillus subtilis* MnmA-YrvO fusion protein can complement *P. falciparum* MnmA. (**A**) Genotyping PCR confirms *mnmA* deletion in *bsmnmA-yrvO⁺ ΔmnmA* parasites. Successful deletion was validated by the presence of PCR amplicons for the Δ5' and Δ3' loci, but not for the unmodified loci (5' and 3') found in the *bsmnmA-yrvO⁺* (parental) parasites. Genotyping PCRs and expected amplicon sizes are described in *Figure 1—figure supplement 1*. (**B**) Mevalonate (Mev)-independent growth of *bsmnmA-yrvO⁺ ΔmnmA* parasites. Asynchronous parasites were grown with or without 50 µM Mev. Every 24 hr parasitemia was monitored by flow cytometry for 4 days. Data points show daily mean parasitemia ± SEM from two independent biological replicates, each

*Figure 5 continued on next page*

*Figure 5 continued*

with four technical replicates; n.s., non-significant, two-way ANOVA (Sidak-Bonferroni method), p>0.05. (**C**) Attempted PCR amplification of *ldh*, *sufB*, and *cox1* genes of the parasite nuclear (N), apicoplast (A), and mitochondrial (M) genomes, respectively, from *bsmnmA-yrvO⁺ ΔmnmA* and *bsmnmA-yrvO⁺* (parental) parasites. Successful amplification of *sufB* in *bsmnmA-yrvO⁺ ΔmnmA* parasites indicates the presence of the apicoplast genome. (**D**) Representative epifluorescence microscopy images of *bsmnmA-yrvO⁺ ΔmnmA* parasites showing an intact apicoplast. (**E**) Growth of the *bsmnmA-yrvO⁺ ΔmnmA* parasites. Asynchronous parasites were grown under permissive (with 0.5 µM aTc and without 0.5 µM *Shield1*) or non-permissive (without 0.5 µM aTc and with 0.5 µM *Shield1*) conditions. Parasitemia was monitored via flow cytometry every 24 hr for 8 days. On day 4, parasite cultures were diluted 1:10. Data points showing daily mean parasitemia ± SEM from two independent biological replicates, each with four technical replicates; two-way ANOVA (Sidak-Bonferroni method), *p<0.05. (**F**) Representative epifluorescence microscopy images of day 8 *bsmnmA-yrvO⁺ ΔmnmA* parasites from (**E**). The apicoplast remains intact under permissive conditions, whereas multiple discrete vesicles were observed under non-permissive conditions. The *Bs* MnmA-YrvO-mCherry (red) is only visible under permissive conditions. The full image panel is available in *Figure 5—figure supplement 1*. In (**D**) and (**F**), the *Bs* MnmA-YrvO fusion protein contains a C-terminal mCherry fluorescent protein (red). Api-SFG protein (green) labels the apicoplast, and nuclear DNA is stained with DAPI (blue). Each image depicts a field of 10 µm × 10 µm. In (**A**) and (**C**), DNA markers are in kilobases (kb).

The online version of this article includes the following source data and figure supplement(s) for figure 5:

**Source data 1.** Uncropped agarose gel images of PCR analyses presented in *Figure 5A and C*.

**Source data 2.** *bsmnmA-yrvO⁺ΔmnmA* growth assay parasitemia counts used for *Figure 5B*.

**Source data 3.** *bsmnmA-yrvO⁺ΔmnmA* growth assay parasitemia counts used for *Figure 5E*.

**Figure supplement 1.** Live epifluorescence microscopy of day 8 samples from permissive and non-permissive growth conditions for *bsmnmA-yrvO⁺ ΔmnmA*.

possessed intact apicoplasts (*Figure 5C and D*), suggesting complementation of *Pf* MnmA activity with *Bs* MnmA-YrvO fusion protein activity. Additionally, the expression and apicoplast localization of *Bs* MnmA-YrvO fusion protein was also confirmed via live epifluorescence microscopy (*Figure 5D*).

To demonstrate complementation of *Pf* MnmA with *Bs* MnmA-YrvO fusion protein more conclusively, we next attempted to knockdown *Bs* MnmA-YrvO in the *bsmnmA-yrvO⁺ ΔmnmA* parasite line by utilizing the TetR-DOZI and CLD systems (*Figure 4A*). We monitored the growth of *bsmnmA-yrvO⁺ ΔmnmA* parasites under permissive (aTc added, *Shield1* removed) and non-permissive conditions (aTc removed, *Shield1* added) for 8 days. Under the non-permissive condition, the *bsmnmA-yrvO⁺ ΔmnmA* parasites showed a significant growth defect from day 4 onward (*Figure 5E*). Live epifluorescence microscopy on day 8 revealed that parasites grown under the non-permissive condition exhibited a disrupted apicoplast phenotype (*Figure 5F*, *Figure 5—figure supplement 1*). This conclusively shows that the *Bs* MnmA-YrvO fusion protein is required to maintain the integrity of the apicoplast organelle after deletion of *Pf* MnmA.

## *Bacillus subtilis* MnmA alone can complement the loss of parasite MnmA

The results from the previous section showed that *Pf* MnmA can be complemented by the *Bs* MnmA-YrvO fusion protein. However, whether *Bs* MnmA alone can complement *Pf* MnmA, without its cognate cysteine desulfurase, *Bs* YrvO, was not clear. To this end, we attempted to delete *pfmnmA* in the *bsmnmA⁺* line. Only one out of four attempts to generate a *bsmnmA⁺ ΔmnmA* line was successful (*Figure 6A*, *Supplementary file 1*b). These parasites were mevalonate independent (*Figure 6B*) and possessed intact apicoplasts (*Figure 6C, D*), replicating the same phenotype as observed for *Bs* MnmA-YrvO complementation (*Figure 5B, C and D*). These results suggested that *Bs* MnmA alone was successful in complementing *Pf* MnmA. Next, we attempted to knockdown *Bs* MnmA in the *bsmnmA⁺ ΔmnmA* parasite line using the TetR-DOZI and CLD systems. We monitored the growth of *bsmnmA⁺ ΔmnmA* parasites under permissive and non-permissive conditions for 8 days. These parasites showed significant growth inhibition under the non-permissive condition from day 3 onward (*Figure 6E*). Live epifluorescence microscopy of parasites cultured under the non-permissive condition showed a disrupted apicoplast on day 8 (*Figure 6F*, *Figure 6—figure supplement 1*). These results conclusively show that complementation of *ΔmnmA* parasites with *Bs* MnmA can maintain

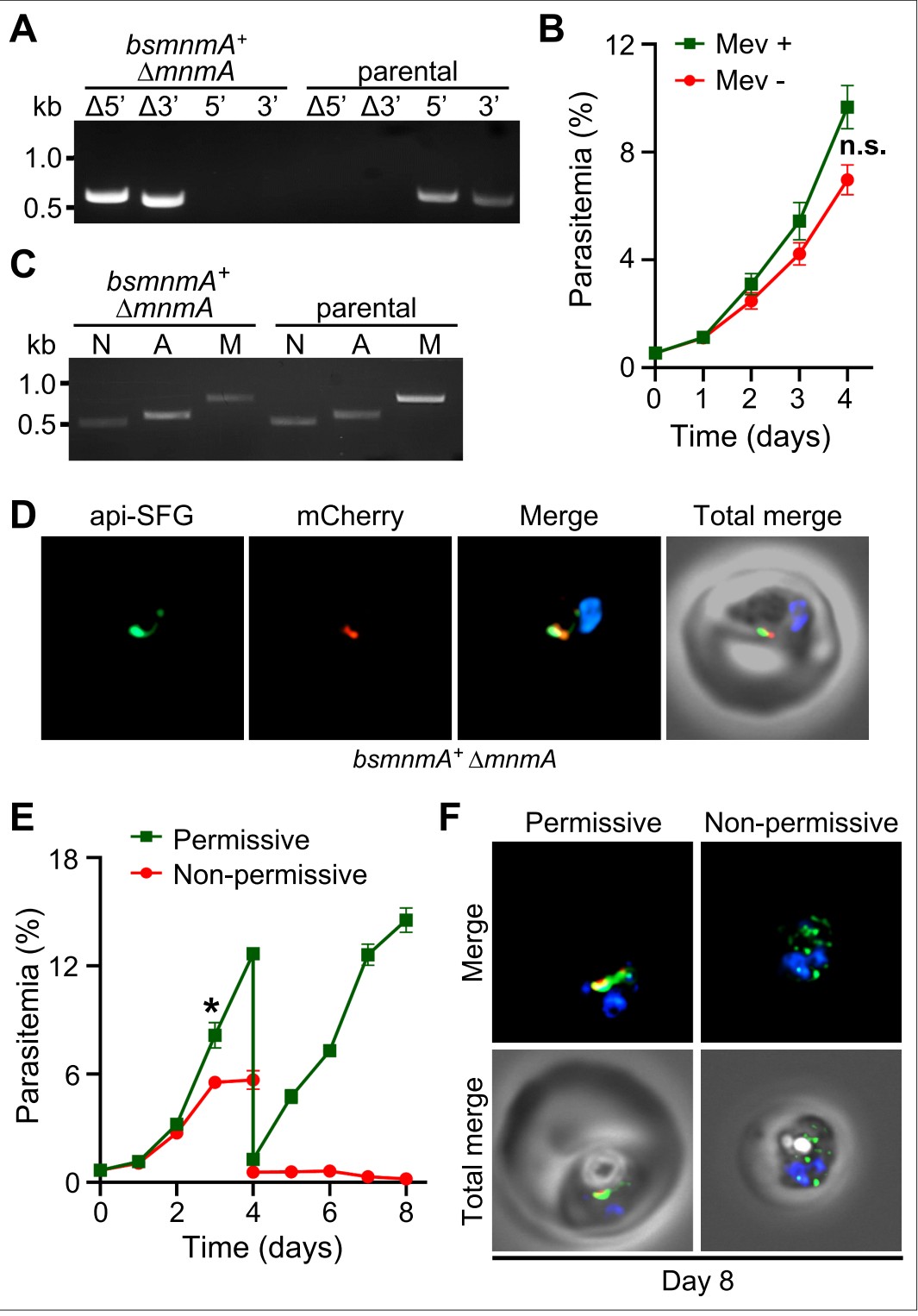

**Figure 6.** *Bacillus subtilis* MnmA can complement loss of *P. falciparum* MnmA. (**A**) Genotyping PCR confirms *mnmA* deletion in *bsmnmA⁺ ΔmnmA* parasites, as evidenced by the presence of PCR amplicons for the Δ5' and Δ3' loci, but not for the unmodified loci (5' and 3') found in the *bsmnmA⁺* (parental) parasites. Genotyping PCRs and expected amplicon sizes are described in *Figure 1—figure supplement 1*. (**B**) Mevalonate (Mev)-independent growth of *bsmnmA ⁺ ΔmnmA* parasites. Asynchronous parasites were grown with or without 50 μM Mev and parasitemia was monitored every 24 hr by flow cytometry for 4 days. Data points represent daily mean parasitemia ± SEM from two independent biological replicates, each with four technical replicates; n.s., non-significant, two-

*Figure 6 continued*

way ANOVA (Sidak-Bonferroni method), p>0.05. (**C**) PCR detection of *ldh*, *sufB*, and *cox1* genes of the parasite nuclear (N), apicoplast (A), and mitochondrial (M) genomes, respectively, in *bsmnmA⁺ ΔmnmA* and *bsmnmA⁺* (parental) parasites. Successful amplification of *sufB* in the *bsmnmA⁺ ΔmnmA* parasites indicates the presence of the apicoplast genome. (**D**) Representative epifluorescence microscopy images of *bsmnmA⁺ ΔmnmA* parasites shows an intact apicoplast. (**E**) Asynchronous *bsmnmA⁺ ΔmnmA* parasites were grown under permissive (with 0.5 µM aTc and without 0.5 µM *Shield1*) or non-permissive (without 0.5 µM aTc and with 0.5 µM *Shield1*) conditions. Parasitemia was monitored via flow cytometry every 24 hr for 8 days. On day 4, parasite cultures were diluted 1:10. Data points represent daily mean parasitemia ± SEM from two independent biological replicates, each with four technical replicates; two-way ANOVA (Sidak-Bonferroni method), *p<0.05. (**F**) Representative epifluorescence microscopy images of day 8 *bsmnmA⁺ ΔmnmA* parasites from (**E**). The apicoplast remained intact under permissive conditions, while multiple discrete vesicles were observed under non-permissive conditions, indicative of a disrupted apicoplast organelle. The *Bs* MnmA -mCherry (red) is only visible under permissive conditions. The full image is available in ***Figure 6—figure supplement 1***. In (**D**) and (**F**), The Bs MnmA protein is tagged C-terminally with mCherry (red), api-SFG protein (green) labels the apicoplast, and nuclear DNA is stained with DAPI (blue). Each image depicts a field of 10 µm × 10 µm. In (**A**) and (**C**), DNA markers are in kilobases (kb).

The online version of this article includes the following source data and figure supplement(s) for figure 6:

**Source data 1.** Uncropped agarose gel images of PCR analyses presented in ***Figure 6A and C***.

**Source data 2.** *bsmnmA⁺ ΔmnmA* growth assay parasitemia counts used for ***Figure 6B***.

**Source data 3.** *bsmnmA⁺ ΔmnmA* growth assay parasitemia counts used for ***Figure 6E***.

**Figure supplement 1.** Live epifluorescence microscopy of day 8 samples from permissive and non-permissive growth conditions for *bsmnmA ⁺ ΔmnmA* parasites.

**Figure supplement 2.** Growth comparison between the *bsmnmA⁺ ΔmnmA* parasite line and *bsmnmA-yrvO⁺ ΔmnmA* parasite line.

**Figure supplement 2—source data 1.** Growth assay parasitemia counts used for ***Figure 6—figure supplement 2***.

the integrity of the organelle. These results also suggest that *Bs* MnmA is likely capable of receiving sulfur from the endogenous parasite cysteine desulfurase, SufS, which is critical for its function in the thiolation of target tRNAs.

## *Plasmodium falciparum* SufS provides sulfur for both FeS synthesis and tRNA thiolation

Deletion of sufS resulted in a disrupted apicoplast mevalonate-dependent phenotype (***Figure 2***). However, deletion of other SUF pathway components (***Figures 1 and 2***) resulted in an intact apicoplast mevalonate-dependent phenotype. These findings led us to hypothesize that mevalonate dependence results from loss of FeS cofactors needed for isoprenoid synthesis (***Swift et al., 2022***), while apicoplast disruption results from loss of sulfur needed for tRNA modification. Our successful complementation of *Pf* MnmA with *Bs* MnmA suggests that *Pf* MnmA has the same enzymatic activity as the well-characterized bacterial enzyme, and that this activity is essential for parasite survival and apicoplast maintenance (***Figures 5 and 6***). However, the complementation experiments in ***Figures 5 and 6*** did not provide any direct evidence that *Pf* MnmA is reliant on sulfur generated from the endogenous SufS for use in tRNA modification. To probe whether SufS provides sulfur to *Pf* MnmA, we used the *bsmnmA-yrvO⁺* line. In these parasites, we attempted to delete *sufS* with continuous supplementation of mevalonate to determine whether the *Bs* MnmA-YrvO protein can complement the essential activity of SufS. We tested this because not only have *Bs* MnmA and *Bs* YrvO been functionally characterized (***Black and Dos Santos, 2015a***), but because *Pf* SufS and *Bs* YrvO share conserved catalytic residues required for cysteine desulfurase activity (***Figure 7—figure supplement 1***). We expected successful complementation of *Pf* SufS with *Bs* MnmA-YrvO to result in a parasite line with intact apicoplasts and a mevalonate-dependent phenotype (***Figure 7A***), which would suggest that sulfur acquired by *Bs* YrvO is only transferred to MnmA but not to the components of the parasite SUF pathway. We were successful in generating the *bsmnmA-yrvO⁺ ΔsufS* parasite line in the presence of mevalonate (***Figure 7B***). As anticipated, these parasites retained intact apicoplasts as confirmed by both PCR and live epifluorescence microscopy (***Figure 7C and D***). These parasites rely on mevalonate

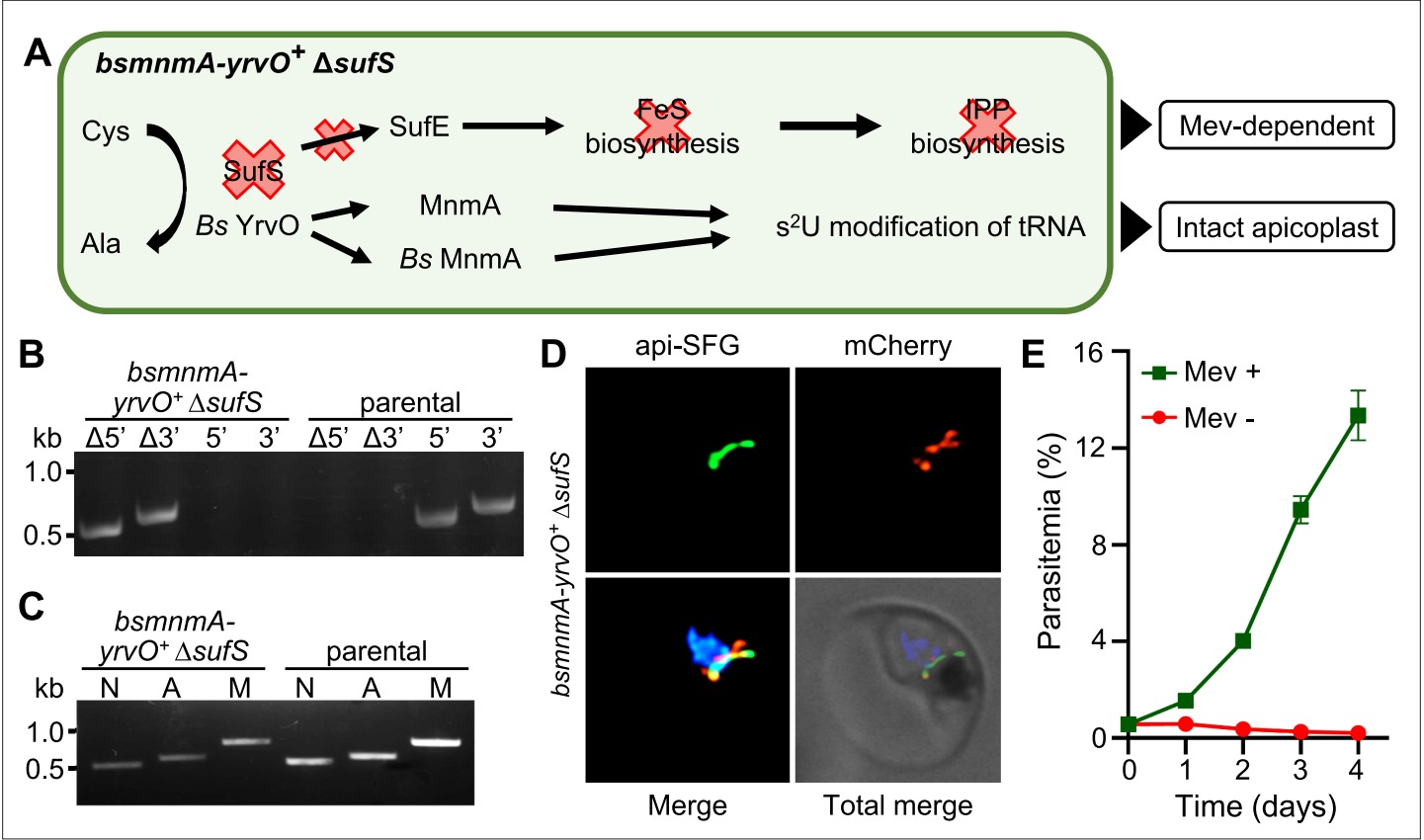

**Figure 7.** *Plasmodium falciparum* SufS provides sulfur for both the sulfur utilization factor (SUF) pathway and MnmA-catalyzed tRNA thiolation. (**A**) The expected outcome of *sufS* deletion in the *bsmnmA-yrvO*+ line is explained in the schematic. In absence of parasite SufS, sulfur for FeS biosynthesis is unavailable resulting in no FeS synthesis. Since the isopentenyl pyrophosphate (IPP) precursor biosynthesis pathway contains FeS-dependent enzymes, this pathway cannot function, rendering parasites reliant on exogenous mevalonate (Mev) to survive. Sulfur liberated by *Bs* YrvO is sufficient for the MnmA-mediated s$^2$U tRNA modification and efficient protein translation. (**B**) Genotyping PCR confirms deletion of *sufS* in *bsmnmA-yrvO*+ *ΔsufS* parasites. Successful deletion was validated by the presence of PCR amplicons for the Δ5′ and Δ3′ loci, but not for the unmodified loci (5′ and 3′) found in the *bsmnmA-yrvO*+ (parental) parasites. Genotyping PCRs and expected amplicon sizes are described in *Figure 1—figure supplement 1*. (**C**) Attempted PCR amplification of *ldh*, *sufB*, and *cox1* genes from the parasite nuclear (N), apicoplast (A), mitochondrial (M) genomes, respectively, in *bsmnmA-yrvO*+ *ΔsufS* and *bsmnmA-yrvO*+ (parental) parasites. Successful amplification of *sufB* in the *bsmnmA-yrvO*+ *ΔsufS* parasite line indicates the presence of the apicoplast genome. (**D**) Representative epifluorescence microscopy images of the *bsmnmA-yrvO*+ *ΔsufS* parasite line expressing api-SFG protein (green) and the *Bs* MnmA-YrvO-mCherry (red). The nuclear DNA is stained with DAPI (blue). A single intact apicoplast was seen in this parasite. Each image depicts a field of 10 μm × 10 μm. (**E**) Mevalonate (mev)-dependent growth of *bsmnmA-yrvO*+ *ΔsufS* parasites. Asynchronous were cultured with or without 50 μM Mev for 4 days. Flow cytometry was used to monitor parasitemia every 24 hr. Data points represent daily mean parasitemia ± SEM from two independent biological replicates, each with four technical replicates. In panels (**B**) and (**C**), DNA markers are in kilobases (kb).

The online version of this article includes the following source data and figure supplement(s) for figure 7:

**Source data 1.** Uncropped agarose gel images of PCR analyses presented in *Figure 7B, C*.

**Source data 2.** *bsmnmA-yrvO*+ *ΔsufS* growth assay parasitemia counts used for *Figure 7E*.

**Figure supplement 1.** Phylogenetic relationship between *P. falciparum* SufS and *B. subtilis* YrvO.

**Figure supplement 1—source data 1.** Multiple sequence alignment of cysteine desulfurase orthologs.

for growth (*Figure 7E*). Collectively, these results strongly suggest that the parasite SufS provides sulfur for both the SUF pathway and MnmA-mediated tRNA modifications.

## Discussion

A recent survey of *P. falciparum* apicoplast proteins found that five proteins are known or predicted to rely on FeS cofactors (*Swift et al., 2022*). Three of these proteins were found to be essential for

the growth of blood-stage parasites due to their roles in supporting the MEP isoprenoid precursor pathway (*Akuh et al., 2022*; *Swift et al., 2022*). These essential proteins could be deleted in PfMev parasites without a noticeable growth defect or loss of the apicoplast organelle as long as the cultures were supplemented with mevalonate. These results implied that the SUF pathway of FeS synthesis would also be essential for parasite growth and that deletion of SUF pathway proteins would not result in loss of the apicoplast. In general, this has proven to be the case with the deletion of SufE, SufC, and SufD. Deletion of SufS, however, led to apicoplast disruption and indicated that this enzyme plays another essential role in parasite biology. Similar to what we observed in *P. falciparum*, deletion of SufS in *T. gondii* also leads to loss of the apicoplast organelle and parasite death (*Pamukcu et al., 2021*). Conditional deletion of SufS in the murine malaria parasite *P. berghei* demonstrated that this enzyme is essential for the development of mosquito-stage parasites, although the status of the apicoplast was not reported in this study (*Charan et al., 2017*). Taken together, these studies suggest that SufS plays a central role in apicoplast biology in other parasite species and other stages of parasite development.

The phenotype of the SufC deletion line contrasts with a previous study (*Gisselberg et al., 2013*) using a dominant negative SufC mutant. In vitro studies showed that *P. falciparum* SufC participates in a SufBC$_2$D complex that hydrolyzes ATP and can form FeS cofactors (*Charan et al., 2017*; *Kumar et al., 2011*). The ATPase activity of SufC is thought to provide energy to drive conformation changes to the entire SufBC$_2$D complex required for iron binding and FeS assembly (*Bai et al., 2018*; *Hirabayashi et al., 2015*; *Yuda et al., 2017*). In *E. coli*, SufC and SufD are essential for SUF pathway activity and acquire iron for FeS assembly; loss of either protein results in reduced iron content in the complex (*Saini et al., 2010*) and the same may be true for the parasite proteins. We generated Δ*sufC*, Δ*sufD*, and Δ*sufC*/*sufD* lines and found that sufC and sufD are essential for parasite survival, but we did not observe an apicoplast disruption phenotype in these deletion lines (*Figure 1*). This finding conflicted with previous results showing that expression of a SufC mutant (K140A) lacking ATPase activity functions as a dominant negative and leads to disruption of the apicoplast (*Gisselberg et al., 2013*). Several factors could be responsible for the apicoplast-disruption phenotype resulting from expression of SufC (K140A). In other organisms, the intact SufBC$_2$D complex enhances SufS desulfurase activity (*Hu et al., 2017a*; *Hu et al., 2017b*; *Outten et al., 2003*; *Wollers et al., 2010*) potentially leading to accumulation of toxic S$^{-2}$ if non-functional SufC (K140A) blocks further sulfur utilization. Alternatively, dominant negative SufC could lead to dysfunctional iron homeostasis. ATP binding to SufC elicits a conformational change in the SufBC$_2$D complex, exposing sites required for iron binding and enabling the formation of nascent clusters (*Bai et al., 2018*; *Hirabayashi et al., 2015*; *Yuda et al., 2017*). The dominant negative mutant SufC should be able to bind ATP, but not hydrolyze it, locking the SufBC$_2$D complex in an open position and exposing these sites to the environment. Exposure and release of iron could lead to oxidative damage and loss of the organelle.

Gene deletion studies exposed different roles for the two proteins (SufE and SufS) involved in sulfur acquisition. We found that both SufE and SufS are required for parasite survival, however, only SufS is required for apicoplast maintenance (*Figures 2, 8A, B and C*). These phenotypes make sense if SufE is required for FeS synthesis but not for tRNA thiolation. In other organisms, SufE has been shown to be capable of enhancing the cysteine desulfurase activity of SufS (*Nmu et al., 2007*; *Outten et al., 2003*; *Pilon-Smits et al., 2002*; *Wollers et al., 2010*; *Ye et al., 2006*) and this appears to be the case with the *P. falciparum* proteins with ~17-fold rate enhancement reported (*Charan et al., 2014*). SufE proteins also facilitate the transfer of sulfur to the SufBC$_2$D complex, but SufS enzymes can often transfer sulfur directly to the complex as is presumably the case in organisms lacking SufE (*Huet et al., 2005*). The requirement for SufE in *P. falciparum* suggests that either SufS desulfurase activity or SufS sulfur transfer activity (to the SufBC$_2$D complex) is too low in the absence of SufE to support FeS synthesis. The fact that SufE is not required for apicoplast maintenance (and presumably tRNA thiolation) further suggests that SufS desulfurase activity in the absence of SufE is adequate for tRNA thiolation and that SufS may transfer sulfur directly to MnmA.

A recent study in the asexual blood-stage of *P. falciparum* reported 28 different tRNA modifications, including s$^2$U and mcm$^5$s$^2$U (*Ng et al., 2018*). The s$^2$U modification of tRNA is ubiquitous and critical for a number of biological functions related to protein translation as demonstrated in other organisms, including the recognition of wobble codons (*Urbonavicius et al., 2001*), tRNA ribosome binding (*Ashraf et al., 1999*), reading frame maintenance (*Black and Dos Santos, 2015a*), and the

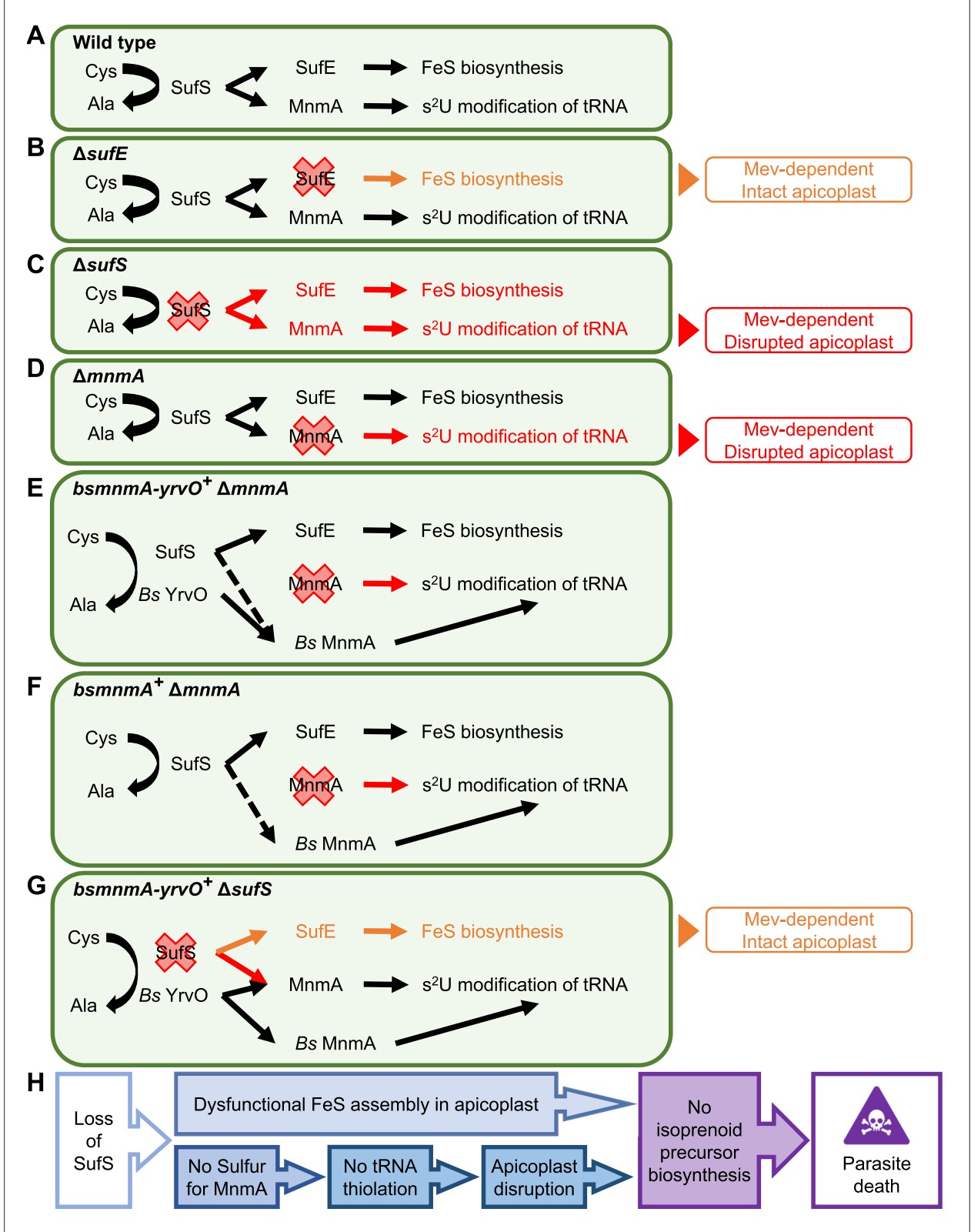

**Figure 8.** Dual role of SufS in the *P. falciparum* apicoplast. (**A**) Dual role of SufS in FeS synthesis and s²U modification of tRNAs in the apicoplast of *P. falciparum*. (**B**) In Δ*sufE* parasites, deletion of *sufE* results in a mevalonate (Mev)-dependent phenotype with an intact apicoplast. This phenotype is likely caused by the deletion of SufE resulting in the loss of FeS biosynthesis. (**C**) In Δ*sufS* parasites, disruption of both FeS biosynthesis and MnmA-dependent tRNA thiolation results in a mevalonate (Mev)-dependent phenotype with a disrupted apicoplast. (**D**) In Δ*mnmA* parasites, deletion of *mnmA*

*Figure 8 continued on next page*

*Figure 8 continued*

results in a mevalonate (Mev)-dependent phenotype with a disrupted apicoplast. Deletion of MnmA likely causes tRNA thiolation defect leading to translational aberration and ultimately apicoplast disruption. (**E**) In *bsmnmA-yrvO⁺ ΔmnmA* parasites, both SufS and *Bs* YrvO have cysteine desulfurase activity that transfers sulfur to *Bs* MnmA in the absence of the endogenous parasite MnmA. (**F**) In *bsmnmA⁺ ΔmnmA* parasites, *Bs* MnmA complements the loss of the endogenous parasite MnmA. The possibility that SufS can transfer sulfur to *Bs* MnmA is represented by the dotted arrow. (**G**) In *bsmnmA-yrvO⁺ ΔsufS* parasites, *Bs* YrvO can transfer sulfur to *Bs* MnmA and/or parasite MnmA, but not to SufE and the FeS biosynthesis pathway. In this scenario, parasites would have intact apicoplasts due to proper tRNA thiolation but would be Mev-dependent due to loss of FeS biosynthesis. (**H**) Events leading to parasite death because of loss of SufS activity in the apicoplast. In panels, (**A–G**), Cys, cysteine; Ala, alanine.

reduction of +1 and +2 frameshifts (***Black and Dos Santos, 2015a***; ***Urbonavicius et al., 2001***). In *P. falciparum*, NCS2 (PF3D7_1441000) is predicted to catalyze s²U modification of nuclear tRNAs while MnmA is predicted to make this modification in the apicoplast. To investigate the biological function of *P. falciparum* MnmA, we complemented parasites lacking endogenous MnmA by expressing the well-studied MnmA ortholog from *B. subtilis* (*Bs* MnmA) with and without its cognate cysteine desulfurase (*Bs* YrvO) partner in the apicoplast. The role of *Bs* MnmA in the s²U modification of target tRNAs has been demonstrated by both genetic and biochemical analysis (***Black and Dos Santos, 2015a***). *Bs* MnmA successfully complemented loss of *P. falciparum* MnmA and resulted in parasites that did not require mevalonate for survival. Subsequent knockdown of the complemented *Bs* MnmA resulted in disruption of the apicoplast, demonstrating that MnmA activity is essential for apicoplast maintenance and parasite survival (***Figures 5, 6, 8E and F***). These results also strongly suggest that *Plasmodium* MnmA catalyzes the thiolation of target tRNAs in the apicoplast. Loss of tRNA thiolation presumably disrupts the translation of essential proteins encoded by the apicoplast genome, ultimately causing the disruption and loss of the organelle. While our complementation assays provide strong support for our claim that *Plasmodium* MnmA has thiolation activity, we could not provide direct biochemical evidence. The fact that knockdown or knockout of MnmA results in apicoplast disruption with subsequent loss of the apicoplast genome (***Figure 3G, H, J and K***) makes it impossible to confirm the loss of any modifications to apicoplast tRNAs. In vitro assays with recombinant MnmA could be another way to demonstrate thiolation activity of MnmA. However, we were unable to construct expression plasmids for MnmA due to the extreme AT richness and low complexity of the gene encoding this enzyme.

For our complementation experiments, we needed multiple attempts to generate a single *bsmnmA⁺ ΔmnmA* line while every attempt to generate *bsmnmA-yrvO⁺ ΔmnmA* parasites was successful (***Supplementary file 1***b). It is possible that a change (indel or mutation) in the *sufS* gene could have facilitated the generation of the single *bsmnmA⁺ ΔmnmA* line. Upon sequencing of the *sufS* gene in this line, we did not observe any changes in the sequence. Another explanation for this phenomenon is that parasite SufS might not be efficient in providing sulfur to *Bs* MnmA (***Figure 8E***). By contrast, *Bs* YrvO can effectively transfer sulfur to *Bs* MnmA in *bsmnmA-yrvO⁺ ΔmnmA* parasites (***Figure 8F***). Despite our difficulties in obtaining *ΔmnmA* parasites in the *bsmnmA⁺* background, we did not observe a statistically significant growth difference between *bsmnmA⁺ ΔmnmA* and *bsmnmA-yrvO⁺ ΔmnmA* parasite lines (***Figure 6—figure supplement 2***). The specificity of interaction between *Bs* MnmA and desulfurase partners makes sense considering the specificity observed in *B. subtilis*. Even though *B. subtilis* produces four SufS paralogs, only YrvO appears to function with *Bs* MnmA (***Black and Dos Santos, 2015a***) and both proteins are essential for bacterial growth (***Kobayashi et al., 2003***).

To show that parasite MnmA receives sulfur from the endogenous cysteine desulfurase, SufS, we deleted SufS in the *bsmnmA-yrvO⁺* line. This deletion resulted in parasites that were reliant on mevalonate for survival but contained an intact apicoplast (***Figure 7***). Mevalonate dependency in *bsmnmA-yrvO⁺ ΔsufS* parasites can be explained by the likely lack of interaction between *Bs* YrvO and other SUF pathway proteins. If *Bs* YrvO cannot transfer sulfur to SufE or to the SufBC₂D complex directly, FeS cofactors will not be synthesized in the apicoplast. Recent results show that lack of FeS cofactors generated by the SUF pathway prevents essential isoprenoid precursor synthesis leading to parasite death in absence of an exogenous source of isoprenoid precursors (***Swift et al., 2022***). Presumably, the apicoplast remains intact in the *bsmnmA-yrvO⁺ ΔsufS* line because *Bs* YrvO and *Bs* MnmA (and *Pf* MnmA) function together to produce essential tRNA s²U modifications (***Figures 7 and 8G***). Taken together, these results show that the parasite SufS provides sulfur for both FeS biosynthesis and

the MnmA-mediated $s^2U$ modification of target tRNAs in the parasite apicoplast, both of which are required for parasite survival (*Figure 8A and C*). Although loss of FeS biosynthesis is not required for apicoplast maintenance, tRNA modification is. Thus, deletion of SufS results in loss of FeS biosynthesis and tRNA modification in the apicoplast, culminating in apicoplast disruption and parasite death (*Figure 8H*).

The dual activity of *P. falciparum* SufS and its direct interaction with MnmA are not typical features of SufS desulfurases. In *E. coli*, IscS is required for MnmA-mediated tRNA thiolation and the role of SufS is confined exclusively to FeS biosynthesis (*Buhning et al., 2017*). Thiolation of tRNA in *E. coli* is also an indirect process involving the five proteins of the Tus system to transfer sulfur from the desulfurase to MnmA (*Black and Dos Santos, 2015a*; *Ikeuchi et al., 2006*; *Shigi, 2014*; *Shigi, 2018*). Similarly, the cysteine desulfurase of *S. cerevisiae* (called Nfs1) is an IscS ortholog and accomplishes the same feat through four sulfur transferases (*Nakai et al., 2004*; *Shigi, 2018*). While homology searches failed to detect intermediate sulfur transferases in *P. falciparum*, a detailed bioinformatic analysis of the 421 aa long N-terminal end of *Pf* MnmA indicated the presence of a SufE-like structural domain (*Figure 3—figure supplement 8*). This putative SufE-like domain could potentially receive the sulfur from *Pf* SufS, and subsequently transfer the sulfur to the C-terminal MnmA domain for tRNA thiolation. Nevertheless, the tRNA thiolation mechanism found in *B. subtilis* is most similar to what we have observed in *P. falciparum*; a SufS paralog (YrvO) transfers sulfur directly to MnmA (*Black and Dos Santos, 2015a*). The unique feature of the *P. falciparum* system is that SufS has dual activity and is essential for two metabolic pathways, while *B. subtilis* YrvO is dedicated to tRNA thiolation (*Black and Dos Santos, 2015a*). We could not identify any conserved motif(s) between *Pf* SufS and *Bs* YrvO which would be indicative of their ability to transfer sulfur directly to their respective MnmA proteins. In fact, we found that these proteins represent two different classes of cysteine desulfurase (*Figure 7—figure supplement 1*). Further studies are necessary to understand what allows *Pf* SufS (and *Bs* YrvO) able to transfer sulfur directly.

In this work and a previous publication (*Swift et al., 2022*), we defined the roles of the five known FeS-dependent proteins and six of the seven proteins involved in the SUF FeS synthesis pathway (SufB was not studied because we do not have a way to target the apicoplast genome). The combined information supports a model in which SufS is needed for both FeS biosynthesis and tRNA thiolation in the apicoplast. In blood-stage parasites, the SUF pathway is required solely for providing FeS cofactors to enable isoprenoid precursor synthesis, but FeS cofactors should also be required for lipoic acid and fatty acid biosynthesis in mosquito-stage and liver-stage parasites (*Akuh et al., 2022*; *Shears et al., 2015*). All stages of parasite development may require tRNA modifications due to the role they play in codon recognition (*Urbonavicius et al., 2001*) which may have elevated importance in the apicoplast due to the unusually small number of only 24 tRNAs (*Wilson et al., 1996*). The dual roles of SufS may be a feature of the apicoplasts found in other pathogens such as *T. gondii*. Deletion of *T. gondii* SufS or MnmA results in apicoplast defects and parasite death (*Pamukcu et al., 2021*; *Yang et al., 2022*). Overall, the work presented here reveals a novel metabolic paradigm and exposes new vulnerabilities in malaria parasites that may extend to other related apicomplexan parasites.

## Materials and methods
### *Plasmodium falciparum* parental parasite lines

For generating the knockout and knock-in lines we used two different parental lines: PfMev (*Swift et al., 2020*) and PfMev^attB (*Swift et al., 2021*). Both parasite lines can generate isoprenoid precursors from an engineered cytosolic mevalonate-dependent pathway (*Swift et al., 2020*; *Swift et al., 2021*). In the presence of mevalonate, these parasite lines can replicate normally even in the absence of an intact apicoplast. The PfMev^attB parasite line has an attB site in the P230p locus (*Swift et al., 2021*). In both parasite lines, the apicoplast is labeled by the super-folder green fluorescent protein (api-SFG) which has been codon-modified for expression in *P. falciparum* (*Roberts et al., 2019*). The api-SFG reporter contains the signal and transit peptide (first 55 aa) of the *P. falciparum* ACP appended to the N-terminal end of SFG to direct trafficking to the apicoplast (*Swift et al., 2020*; *Swift et al., 2021*).

### *Plasmodium falciparum* culture

Asexual stage *P. falciparum* parasites were cultured in human O$^+$ erythrocytes at 2% hematocrit in RPMI 1640 medium with L-glutamine (USBiological, MA, USA). The RPMI 1640 medium was supplemented with 20 mM HEPES, 0.2% sodium bicarbonate, 12.5 μg/mL hypoxanthine, 5 g/L Albumax II (Thermo Fisher Scientific, MA, USA), and 25 μg/mL gentamicin. Cultures were incubated at 37°C and maintained in 25 cm$^2$ gassed flasks (94% $N_2$, 3% $O_2$, and 3% $CO_2$).

## Generation of *P. falciparum* plasmid constructs for gene deletion

We employed Cas9-mediated gene editing to delete genes of interest (*Figure 1—figure supplement 1*). For gene deletion, the pRS repair plasmid (*Swift et al., 2020*) in combination with the pUF1-Cas9 plasmid (*Ghorbal et al., 2014*) were used. Alternatively, the pRSng (*Swift et al., 2020*) or pRSng(BSD) repair plasmid (*Swift et al., 2022*) in combination with the pCasG-LacZ plasmid (*Rajaram et al., 2020*) were used for targeted gene deletion. The plasmids used for generation of the gene deletion lines are listed in *Supplementary file 1*c.

To generate deletion constructs, ~300–500 bp homology arms (HAs) were amplified from *P. falciparum* NF54 genomic DNA with HA1 and HA2 forward and reverse primers (*Supplementary file 1*d). The HA1 and HA2 amplicons were inserted into the *NotI* and *NgoMIV* restriction sites, respectively, of the repair plasmids by In-Fusion (Clontech Laboratories, CA, USA) ligation independent cloning (LIC). The gRNA sequences (*Supplementary file 1*d) were inserted as annealed oligonucleotides into the *BsaI* sites of pCasG-LacZ plasmids by LIC to generate pCasG-GOI plasmid or into the pRS plasmid. All the plasmids were sequenced to confirm sequence fidelity. All the restriction enzymes used in this section were sourced from New England Biolabs Inc, MA, USA.

## Generation of plasmid constructs for knockdown and epitope tagging of *Pf* MnmA

To generate the knockdown construct of the endogenous *Pf* MnmA, we created the pKD-*mnmA*-2xFLAG-10xapt plasmid. This plasmid also allowed us to tag *Pf* MnmA C-terminally with 2xFLAG. To make this plasmid, ~300–400 bp HA1 and HA2 of the *pfmnmA* gene were PCR amplified with MnmAKD.HA1*F*+MnmAKD .HA1R and MnmAKD.HA2*F*+MnmAKD .HA2R, respectively from *P. falciparum* NF54 genomic DNA (*Supplementary file 1*d). These HA PCR fragments were fused together to generate a combined HA2-HA1 fragment with MnmAKD.HA2F and MnmAKD.HA1R in an additional PCR step. The HA2 and HA1 fragments in the combined HA2-HA1 fragment are separated by two *EcoRV* sites to facilitate linearization of the plasmid. The HA2-HA1 fragment was inserted into the *AscI* and *AatII* sites of the pKD-2xFLAG-10xapt plasmid to generate the pKD-*mnmA*-2xFLAG-10xapt plasmid by LIC. The pKD-2xFLAG-10xapt plasmid was generated by replacing the 3xHA tag of the pKD plasmid reported elsewhere (*Rajaram et al., 2020*). To replace the 3xHA tag with the 2xFLAG tag, the FLAG tag sequence was synthesized as oligonucleotides (*Supplementary file 1*d), annealed, and then inserted into the *AatII* and *PspMOI* sites of the pKD plasmid. Prior to transfection, the pKD-*mnmA*-2xFLAG-10xapt plasmid was linearized with *EcoRV*. The MnmA.KD gRNA sequence (*Supplementary file 1*d) was inserted as annealed oligonucleotides into the *BsaI* sites of the pCasG-LacZ plasmid by LIC to generate the pCasG-*mnmA*KD plasmid.

## Generation of plasmid constructs for *B. subtilis mnmA* and *mnmA-yrvO* knock-in

To generate the *bsmnmA* knock-in plasmid, we amplified the *bsmnmA* gene from *B. subtilis* genomic DNA with the following primer pair: MnmA.BspEI.InF.F and MnmA.BsiWI.InF.R. The amplified product was used to replace the *Ec*DPCK locus from the pCLD-*Ec*DPCK-mCherry-apt (*Swift et al., 2021*) plasmid using the *BspEI* and *BsiWI* cloning sites to generate pCLD-*bsmnmA*-mCherry-apt. For enhanced stability of the aptamer system, we replaced the existing 10x-aptamer array (apt) with a redesigned 10x-aptamer array (10xapt) that prevents aptamer loss (*Rajaram et al., 2020*). First, the 10xapt DNA was amplified with the MY.Apt.PspOMI.F and MY.Apt.XmaI.R primer pair from a pKD plasmid (*Rajaram et al., 2020*). The apt locus was removed from the pCLD-*bsmnmA*-mCherry-apt plasmid by digestion with *PspOMI* and *XmaI*, followed by insertion of the 10xapt PCR product using the same cloning sites by LIC resulting in pCLD-*bsmnmA*-mCherry-10xapt plasmid. The fidelity of the *Bs mnmA* and 10xapt sequence of the pCLD-*bsmnmA*-mCherry-10xapt was confirmed by sequencing.

In *B. subtilis*, the genes *mnmA* and *yrvO* are 31 bp apart from each other and are co-transcribed (***Black and Dos Santos, 2015a***). Hence, we decided to generate the pCLD-*bsmnmA-yrvO*-mCherry-10xapt plasmid with *bsmnmA* and *bsyrvO* genes fused in one cassette with a 15 bp linker. To make the fusion gene, we first amplified *mnmA* with MnmA.BspEI.InF.F and MnmA.Link.R, and *yrvO* with YrvO.Link.F and YrvO.BsiWI.InF.R primer pairs. These two PCR products were stitched together by PCR amplification (*bsmnmA-yrvO*, **Figure 4—figure supplement 2**) using the primer pair MnmA.BspEI.InF.F and YrvO.BsiWI.InF.R. The *bsmnmA-yrvO* fragment was inserted into the pCLD-*bsmnmA*-mCherry-10xapt plasmid, previously digested with *BspEI* and *BsiWI*, using LIC. The resulting plasmid, pCLD-*bsmnmA-yrvO*-mCherry-10xapt, was sequenced to confirm the sequence fidelity of *bsmnmA-yrvO*. All the primers are listed in **Supplementary file 1**d and all the restriction enzymes were purchased from New England Biolabs Inc, MA, USA.

## Parasite transfections

To generate the Δ*sufC*, Δ*sufD*, Δ*sufE*, Δ*sufS*, and Δ*mnmA* transgenic lines, PfMev parasites were transfected with the respective plasmids using previously described transfection methods (***Spalding et al., 2010***). Briefly, 75 µg of both gRNA containing pUF1-CasG or pCasG plasmids and corresponding pRS or pRSng repair plasmids (**Supplementary file 1**c) were electroporated into 400 µL of red blood cells (RBCs) by low-voltage electroporation. The transfected RBCs were mixed with 2.5 mL of synchronized schizont-stage PfMev parasites and were cultured in complete medium with 50 µM mevalonate (Racemic mevalonolactone, M4667, MilliporeSigma, MO, USA). After 48 hr, 1.5 µM DSM1 (MRA 1161, BEI resources, VA, USA), 2.5 nM WR99210 (Jacobus Pharmaceuticals, NJ, USA), and 50 µM mevalonate were added to select for the transfectants over the course of 7 days. After 7 days, the cultures were maintained in complete medium with 50 µM mevalonate until parasites appeared in the culture. Once the parasites appeared, the cultures were maintained in complete medium with 2.5 nM WR99210 and 50 µM mevalonate.

For generating the Δ*sufC/sufD* line, RBCs were transfected with the pCasG-*sufC* and pRSng(BSD)-*sufC* plasmids as described above. Synchronized Δ*sufD* parasites were added to the transfected RBCs and cultured in medium with 50 µM mevalonate. The transfectants were selected with 2.5 µg/mL blasticidin (30-100-RB, Corning Inc, NY, USA), 1.5 µM DSM1, 2.5 nM WR99210, and 50 µM mevalonate, after which the culture was maintained in complete medium containing 2.5 nM WR99210 and 50 µM mevalonate until parasites appeared. Upon parasite appearance, the culture was maintained in complete medium with 2.5 µg/mL blasticidin, 2.5 nM WR99210, and 50 µM mevalonate.

For generation of the *mnmA*-flag parasite lines, the linearized pKD-*mnmA*-2xFLAG-10xapt and pCasG-*mnmA*KD plasmids were co-transfected into RBCs as mentioned above. Following transfection, these RBCs were mixed with PfMev[attB] parasites (***Swift et al., 2021***) and selected with 2.5 µg/mL blasticidin and 1.5 µM DSM1 along with 0.5 µM anhydrous tetracycline (aTc, 10009542, Cayman Chemical, MI, USA) for 7 days. After initial selection for 7 days, this culture was grown in complete medium with aTc until parasite reappearance. Upon parasite reappearance, the culture was switched to and maintained in complete medium containing 2.5 µg/mL blasticidin and 0.5 µM aTc.

To generate the *bsmnmA*[+] and *bsmnmA-yrvO*[+] transgenic parasite lines, either the pCLD-*bsmnmA*-mCherry-10xapt or pCLD-*bsmnmA-yrvO*-mCherry-10xapt plasmids were co-transfected into RBCs with the pINT plasmid (***Nkrumah et al., 2006***) encoding the mycobacteriophage integrase (this integrase mediates attP/attB integration into the target genome locus). Transfected RBCs were mixed with PfMev[attB] parasites and cultured with 2.5 µg/mL blasticidin and 0.50 µM aTc for 7 days. After 7 days, these cultures were grown in complete medium with aTc until parasites were observed, at which point the cultures were maintained in complete medium containing 2.5 µg/mL blasticidin and 0.50 µM aTc.

The *bsmnmA*[+] Δ*mnmA*, *bsmnmA-yrvO*[+] Δ*mnmA*, and *bsmnmA-yrvO*[+] Δ*sufS* transgenic parasite lines were generated with the same Cas9 and pRSng repair plasmids that were used to generate the Δ*mnmA* and Δ*sufS* transgenic lines. For *bsmnmA*[+] Δ*mnmA*, *bsmnmA-yrvO*[+] Δ*mnmA*, and *bsmnmA-yrvO*[+] Δ*sufS*, medium supplemented with 1.5 µM DSM1, 2.5 nM WR99210, 1.25 µg/mL blasticidin, and 0.50 µM aTc was used for the initial 7 days of selection, after which the cultures were switched to growth medium containing blasticidin and aTc. Upon parasite appearance, all cultures were maintained in medium containing WR99210, blasticidin, and aTc. The *bsmnmA-yrvO*[+] Δ*sufS* transgenic parasite line was supplemented with 50 µM mevalonate in addition to WR99210, blasticidin, and aTc.

The *bsmnmA*⁺ Δ*mnmA* line was difficult to generate with only one successful line out of four attempts. Between two and eight parasite lines from independent transfections were obtained for all other gene deletions (*Supplementary file 1*b).

## Confirmation of gene knockout, C-terminal tagging, and gene knock-in

Lysates from parasite cultures were prepared from the transgenic parasite lines by incubating at 90°C for 5 min. These lysates were used as the template for all genotype confirmation PCRs. For confirmation of gene knockouts, the 5'- and 3'-end of the disrupted (Δ5' and Δ3', respectively) and native gene loci (5' and 3', respectively) were amplified with corresponding primers (*Supplementary file 1*d). Expected amplicons for confirmation PCRs are provided in *Figure 1—figure supplement 1C*. To confirm the successful C-terminal tagging of MnmA and insertion of the aptamer array at the 3'UTR of MnmA, the 5'- and 3'-end of modified genes (Δ5' and Δ3', respectively) and the native gene locus (C) were amplified with corresponding primers. The expected amplicon sizes for these PCR products are provided in *Figure 3—figure supplement 4B*. For gene knock-in confirmation, corresponding primers (*Supplementary file 1*d) were used to amplify the recombinant attL and attR loci for integration of the knocked-in gene and the unaltered attB locus as a control. The anticipated PCR amplicon sizes for knock-in confirmation PCRs can be found in *Figure 4—figure supplement 1B*. Parental lines as indicated were used as a control for these reactions.

## Immunoblot

Asynchronous parasite cultures were washed three times with cold-complete medium. The cultures were treated with 0.15% (wt/vol) saponin in cold phosphate-buffered saline (PBS; pH 7.4) for 10 min on ice for permeabilization of the RBC and parasitophorous vacuolar membranes. Saponin-isolated parasites were centrifuged at 1940 × *g* for 10 min at 4°C and washed three times with cold PBS. These parasites were used immediately or were snap-frozen in liquid $N_2$ and saved at –80°C for later use.

Saponin-isolated parasites were resuspended in 1× NuPAGE LDS sample buffer (NP0007, Thermo Fisher Scientific, MA, USA) containing 2% β-mercaptoethanol and boiled for 5 min. Proteins were resolved on 4–12% gradient reducing gels and transferred to nitrocellulose membranes. The membranes were blocked with 5% milk in PBS containing 0.1% Tween 20 (milk/PBST) for 1 hr at room temperature and incubated overnight at 4°C with mouse anti-FLAG monoclonal antibody (F3165, MilliporeSigma, MO, USA) diluted 1:1000 in milk/PBST. The membrane was then incubated for 1 hr at room temperature with sheep anti-mouse horseradish peroxidase (HRP)-conjugated antibody (GENA931, MilliporeSigma, MO, USA) diluted 1:10,000 in milk/PBST. Chemiluminescent signal was developed with SuperSignal West Pico chemiluminescent substrate (34577, Thermo Fisher Scientific, MA, USA) according to the manufacturer's protocol and detected on autoradiography film. For loading controls, the membrane was stripped of antibodies with 200 mM glycine (pH 2.0) for 5 min at room temperature. After blocking the membrane with 5% milk/PBST, the membrane was re-probed with 1:25,000 mouse anti-aldolase monoclonal antibody (from David J. Sullivan, Johns Hopkins Bloomberg School of Public Health) and 1:10,000 sheep anti-mouse HRP-conjugated antibody, using the methods described above.

## Confirmation of apicoplast genome loss

The apicoplast-encoded *sufB* gene (PF3D7_API04700) was used as a proxy for the presence of the apicoplast genome and was amplified by PCR with a primer pair listed in *Supplementary file 1*d. As controls, genes from the nuclear genome (*ldh*, PF3D7_1324900) and the mitochondrial genome (*cox1*, PF3D7_MIT02100) were amplified with corresponding primer pairs (*Supplementary file 1*d). Parasite lysates of the parental line were used as positive controls for apicoplast genome detection. The anticipated amplicon sizes are 520 bp, 581 bp, and 761 bp for *ldh*, *sufB,* and *cox1*, respectively.

## Immunofluorescence assays and live cell microscopy

For immunofluorescence assays, *mnmA*-flag parasites were fixed and permeabilized as described previously (*Gallagher et al., 2011*). Briefly, infected RBCs from 250 µL of culture were harvested by centrifugation and resuspended in 30 µL of 4% electron microscopy (EM) grade paraformaldehyde and 0.0075% EM grade glutaraldehyde in PBS (pH 7.4) for 30 min at room temperature. Fixed cells were then permeabilized with 0.1% Triton X-100 in PBS for 10 min and then treated with 0.1 mg/mL of

sodium borohydride (NaBH$_4$) in PBS for 10 min to reduce free aldehyde groups. After a 2 hr blocking step with 3% bovine serum albumin (BSA) in PBS, cells were incubated overnight at 4°C with 1:500 rabbit anti-ACP antibody (*Gallagher and Prigge, 2010*) and 1:500 mouse anti-FLAG monoclonal antibody (F3165, MilliporeSigma, MO, USA). The cells were washed with PBS three times and then incubated for 2 hr with 1:1000 goat anti-mouse Alexa 488 (A11029, Thermo Fisher Scientific, MA, USA) and 1:1000 goat anti-rabbit Alexa 594 (A11037, Thermo Fisher Scientific, MA, USA) secondary antibodies in PBS with 3% BSA. After three washes with PBS, the cells were mounted on coverslips with ProLong Gold 4',6-diamidino-2-phenylindole (DAPI) antifade reagent (P36935, Thermo Fisher Scientific, MA, USA) and sealed with nail polish.

For live cell imaging of the Δ*sufC*, Δ*sufD*, Δ*sufE*, Δ*sufS*, Δ*mnmA*, and Δ*sufC/sufD* transgenic lines, 100 µL of asynchronous parasites were incubated with 1 µg/mL DAPI (62248, Thermo Fisher Scientific, MA, USA) and 30 nM MitoTracker CMX-Ros (M7512, Thermo Fisher Scientific, MA, USA) for 30 min at 37°C. The stained cells were then washed three times with complete medium and incubated for 5 min at 37°C after each wash. After the final wash, the cells were resuspended in 20 µL of complete medium, placed on a slide and sealed under a cover slip with wax. For live cell imaging of the *bsmnmA$^+$*, *bsmnmA-yrvO$^+$*, *bsmnmA$^+$* Δ*mnmA*, *bsmnmA-yrvO$^+$* Δ*mnmA*, and *bsmnmA-yrvO$^+$* Δ*sufS* parasite lines, cells were stained with 1 µg/mL DAPI only.

All images were taken with a Zeiss AxioImager M2 microscope (Carl Zeiss Microscopy, LLC, NY, USA) equipped with a Hamamatsu ORCA-R2 camera (Hamamatsu Photonics, Hamamatsu, Japan) using a 100×/1.4 NA lens. A series of images were taken spanning 5 µm along the z-plane with 0.2 µm spacing. An iterative restoration algorithm using the Volocity software (PerkinElmer, MA, USA) was used to deconvolve the images to report a single image in the z-plane.

The degree of colocalization between the red channel (anti-ACP or mCherry) and green channel (anti-FLAG or api-SFG) signals was determined with Volocity software. The fluorescent intensity thresholds were set using the region of interest tool using Volocity. To set the thresholds, the fluorescence intensity of a region of the cell with no staining for either signal (background) was used. To measure the degree of colocalization, the Manders' coefficient (M1) (*Manders et al., 1993*) was determined. M1 is defined as the percentage of total pixels from the test channel (anti-FLAG, mCherry) that overlaps with the percentage of total pixels from the organellar marker channel (anti-ACP, api-SFG). A value of M1=1 denotes perfect colocalization, while M1=0 denotes no colocalization (*Manders et al., 1993*). Mean M1 (± standard deviation) values were obtained by analyzing multiple images from at least two independent biological replicates.

## Parasite growth assay

Parasite growth was monitored using an Attune Nxt Flow Cytometer (Thermo Fisher Scientific, MA, USA) as previously described (*Swift et al., 2020*; *Tewari et al., 2022*). For determining the growth dependence on mevalonate, cultures were seeded in the presence or absence of 50 µM mevalonate at 0.5% parasitemia and 2% hematocrit in a total volume of 250 µL, in quadruplicate for each condition. Parasite growth was monitored every 24 hr over 4 days following SYBR green I (S7563, Thermo Fisher Scientific, MA, USA) staining. For the growth curves shown in *Figure 3F*, the parasites were cultured in the presence of 0.5 µM aTc (control), absence of aTc (knockdown), or with 50 µM mevalonate supplementation (rescue). For growth assays presented in *Figures 5E and 6E*, parasites were grown in the presence of 0.5 µM aTc and absence of *Shield1* (AOB1848, Aobious Inc, MA, USA) (permissive condition) or in the absence of aTc and presence of 0.5 µM *Shield1* (non-permissive condition). In all the experiments presented in *Figures 3F, 5E and 6E*, parasite growth was monitored over 8 days. On day 4, the cultures were diluted 1:10. Data from two independent biological replicates (each in quadruplicate) of the indicated parasite lines were analyzed using a two-way ANOVA with a Sidak-Bonferroni correction in Prism V8.4 (GraphPad Software, CA, USA).

## MSA and structural alignments

All of the MSA were generated with the ClustalW program in MEGA11 (*Tamura et al., 2021*). The MSA used for *Figure 3A*, *Figure 3—figure supplements 1 and 2*, and *Figure 7—figure supplement 1* were done with following full-length proteins: *P. falciparum* (*Pf* MnmA, PlasmoDB ID: PF3D7_1019800), *Synechococcus* sp. (*Sy* MnmA, Uniprot ID: Q2JY34), *B. subtilis* (*Bs* MnmA, Uniprot ID: O35020), *E. coli* MnmA (*Ec* MnmA, Uniprot ID: P25745), *A. thaliana* (*At* MnmA, Uniprot ID: Q9SYD2), *S. cerevisiae*

(*Sc* Mtu1, Uniprot ID: Q12093), *B. microti* (PiroplasmaDB ID: BMR1_03g03155), *T. annulata* (PiroplasmaDB ID: TA12620), *T. gondii* (ToxoDB ID: TGME49_309110), *E. tenella* (ToxoDB ID: ETH_00027940), *P. falciparum* (*Pf* SufS, PlasmoDB ID: PF3D7_0716600), *B. subtilis* (*Bs* SufS, Uniprot ID: O32164; *Bs* YrvO, Uniprot ID: O34599), *E. coli* (*Ec* SufS, Uniprot ID: P77444; *Ec* IscS, Uniprot ID: P0A6B7), *A. thaliana* (*At* cNif1, Uniprot ID: Q93WX6; *At* mNif1, Uniprot ID: O49543), and *S. cerevisiae* (*Sc* Nfs1, Uniprot ID: P25374).

Structural alignments presented in *Figure 3—figure supplement 3* were done with these regions of the following proteins: *Synechococcus* sp. (*Sy* MnmA), *B. subtilis* (*Bs* MnmA), *E. coli* MnmA (*Ec* MnmA), *A. thaliana* (*At* MnmA), *S. cerevisiae* (*Sc* Mtu1), *P. falciparum* MnmA (*Pf* MnmA, residues 422–1083), *B. microti* (*Bm* MnmA, residues 235–680), *T. annulata* (*Ta* MnmA, residues 273–1070), *T. gondii* (*Tg* MnmA, residues 807–1598), and *E. tenella* (*Et* MnmA, residues 445–974). For *Figure 3—figure supplement 8*, the following proteins were used: *E. coli* SufE (*Ec* SufE, Uniprot ID: P76194), *P. falciparum* SufE (*Pf* SufE, residues 101–249; PlasmoDB ID: PF3D7_0206100), *P. falciparum* MnmA (*Pf* MnmA, residues 146–380), *B. microti* (*Bm* MnmA, residues 85–224), *T. annulata* (*Ta* MnmA, residues 117–272), *T. gondii* (*Tg* MnmA, residues 337–708), and *E. tenella* (*Et* MnmA, residues 145–444).

## Predicted structure and phylogenetic analysis

The crystal structure of *E. coli* SufE (PDB id: 1MZG) and *E. coli* MnmA (PDB id: 2DER) were obtained from RCSB PDB (https://www.rcsb.org/) (*Burley et al., 2020*) and used as templates for generating homology models. Homology models of *P. falciparum* SufE and MnmA, and other orthologs of SufE and MnmA were predicted with Phyre2 protein fold recognition server (https://www.sbg.bio.ic.ac.uk/servers/phyre2) (*Kelley et al., 2015*) using the '*intensive*' modeling mode. In all cases, the top predicted homology models were used for further analysis and contained the following regions: *Pf* MnmA C-terminal domain (Arg449-Asn704 and Asp721-Tyr1017), *Pf* MnmA N-terminal domain (Ile146-Tyr380), and *Pf* SufE (Leu105-Ile243). The crystal structures and the predicted models were visualized with UCSF ChimeraX (version: 1.3rc202111110135; https://www.rbvi.ucsf.edu/chimerax) molecular visualization program (*Pettersen et al., 2021*). Structural alignments were generated with the '*Matchmaker*' tool of UCSF ChimeraX. Phylogenetic analysis was done with MEGA11 (*Tamura et al., 2021*) using the maximum likelihood method and the JTT matrix-based model (*Jones et al., 1992*). In all cases 1000 replicates of bootstrap analyses were performed and the tree with the highest likelihood is presented.

## Amplicon sequencing of *P. falciparum* SufS

The gene encoding *P. falciparum* SufS (PF3D7_0716600) was PCR amplified in three overlapping fragments with Phusion DNA polymerase (New England Biolabs Inc, MA, USA) using the primers in *Supplementary file 1*d. PCR amplicons were sequenced with the primers in *Supplementary file 1*d and aligned with the '*blastn*' tool of the Basic local alignment search tool (https://blast.ncbi.nlm.nih.gov/Blast.cgi) (*Altschul et al., 1990*).

## Acknowledgements

We express our gratitude to Patricia C Dos Santos (Wake Forest University) for insights on *B. subtilis* MnmA and YrvO. We also extend our thanks to David J Sullivan (Johns Hopkins Bloomberg School of Public Health) for the mouse anti-aldolase monoclonal antibody and Erin D. Goley (Johns Hopkins University School of Medicine) for providing *B. subtilis*.

This work was supported by the National Institutes of Health R01 AI125534 (STP) and R21 AI101589 (STP), the Johns Hopkins Malaria Research Institute, and the Bloomberg Philanthropies. RE was supported by Johns Hopkins Malaria Research Institute postdoctoral fellowship. KR was supported by NIH training grant T32AI007417. The funders had no role in study design, data collection and analysis, decision to publish, or preparation of the manuscript.

# Additional information

## Funding

| Funder | Grant reference number | Author |
| --- | --- | --- |
| Johns Hopkins Malaria Research Institute, Johns Hopkins Bloomberg School of Public Health | Postdoctoral fellowship | Rubayet Elahi |
| National Institute of Allergy and Infectious Diseases | R01AI125534 | Sean T Prigge |
| National Institute of Allergy and Infectious Diseases | R21AI101589 | Sean T Prigge |
| National Institute of Allergy and Infectious Diseases | T32AI007417 | Krithika Rajaram |

The funders had no role in study design, data collection and interpretation, or the decision to submit the work for publication.

## Author contributions

Russell P Swift, Rubayet Elahi, Conceptualization, Data curation, Formal analysis, Validation, Investigation, Visualization, Methodology, Writing – original draft, Writing – review and editing; Krithika Rajaram, Data curation, Formal analysis, Validation, Investigation, Visualization, Methodology, Writing – review and editing; Hans B Liu, Data curation, Validation, Investigation, Visualization, Methodology, Writing – review and editing; Sean T Prigge, Conceptualization, Resources, Supervision, Funding acquisition, Methodology, Writing – original draft, Project administration, Writing – review and editing

## Author ORCIDs

Rubayet Elahi ⬚ http://orcid.org/0000-0002-1561-5257
Krithika Rajaram ⬚ http://orcid.org/0000-0003-4830-5471
Sean T Prigge ⬚ http://orcid.org/0000-0001-9684-1733

## Decision letter and Author response

Decision letter https://doi.org/10.7554/eLife.84491.sa1
Author response https://doi.org/10.7554/eLife.84491.sa2

# Additional files

## Supplementary files

• Supplementary file 1. Supplementary information for reproducing and interpreting data presented in the manuscript. (a) Sulfur-dependent pathways present in *P. falciparum*. (b) Summary of attempted transfections in *P. falciparum* to knockout genes of interest. (c) Plasmids used for generation of gene deletion lines. (d) Primers used in this study. Restriction enzyme sites are single underlined, linker regions are double underlined.

• MDAR checklist

## Data availability

All data generated or analyzed during this study are included in the manuscript and supporting file; Source Data files have been provided for Figures 1, 2, 3, 4, 5, 6, and 7.

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

# Appendix 1

## Appendix 1—key resources table

| Reagent type (species) or resource | Designation | Source or reference | Identifiers | Additional information |
|---|---|---|---|---|
| Gene (*Plasmodium falciparum*) | SufC | PlasmoDB (https://plasmodb.org) | PF3D7_1413500 | *P. falciparum* SufC gene |
| Gene (*P. falciparum*) | SufD | PlasmoDB (https://plasmodb.org) | PF3D7_1103400 | *P. falciparum* SufD gene |
| Gene (*P. falciparum*) | SufE | PlasmoDB (https://plasmodb.org) | PF3D7_0206100 | *P. falciparum* SufE gene |
| Gene (*P. falciparum*) | SufS | PlasmoDB (https://plasmodb.org) | PF3D7_0716600 | *P. falciparum* SufS gene |
| Gene (*P. falciparum*) | MnmA | PlasmoDB (https://plasmodb.org) | PF3D7_1019800 | *P. falciparum* MnmA gene |
| Cell line (*P. falciparum*) | PfMev | *Swift et al., 2020* https://doi.org/10.1371/journal.ppat.1008316 | | Parasites with engineered cytosolic mevalonate pathway. Available from the Prigge laboratory, Johns Hopkins Bloomberg School of Public Health |
| Cell line (*P. falciparum*) | PfMev^attB | *Swift et al., 2021* https://doi.org/10.15252/embj.2020107247 | | PfMev parasites with an attB site in the P230p locus. Available from the Prigge lab |
| Genetic reagent (*P. falciparum*) | ΔsufC | This paper | | *sufC* gene deletion line in PfMev parental background. Available from the Prigge lab |
| Genetic reagent (*P. falciparum*) | ΔsufD | This paper | | *sufD* gene deletion line in PfMev parental background. Available from the Prigge lab |
| Genetic reagent (*P. falciparum*) | ΔsufE | This paper | | *sufE* gene deletion line in PfMev parental background. Available from the Prigge lab |
| Genetic reagent (*P. falciparum*) | ΔsufS | This paper | | *sufS* gene deletion line in PfMev parental background. Available from the Prigge lab |
| Genetic reagent (*P. falciparum*) | ΔmnmA | This paper | | *mnmA* gene deletion line in PfMev parental background. Available from the Prigge lab |
| Genetic reagent (*P. falciparum*) | ΔsufC/sufD | This paper | | *sufC* and *sufD* double gene deletion line in PfMev parental background. Available from the Prigge lab |
| Genetic reagent (*P. falciparum*) | mnmA-flag | This paper | | Parasites expressing C-terminally FLAG tagged endogenous *Pf* MnmA in the PfMev^attB parental background. An aptamer array in the 3' untranslated region (UTR) of *mnmA* allows inducible control. Available from the Prigge lab |
| Genetic reagent (*P. falciparum*) | bsmnmA^+ | This paper | | Parasites expressing the *B. subtilis* MnmA-mCherry fusion protein in the PfMev^attB parental background. An aptamer array in the 3' UTR of *bsmnmA-mCherry* allows inducible control. Available from the Prigge lab |
| Genetic reagent (*P. falciparum*) | bsmnmA-yrvO^+ | This paper | | Parasites expressing the *B. subtilis* MnmAYrvO-mCherry fusion in the PfMev^attB parental background. An aptamer array in the 3' UTR of *bsmnmAyrvO-mCherry* allows inducible control. Available from the Prigge lab |
| Genetic reagent (*P. falciparum*) | bsmnmA^+ ΔmnmA | This paper | | *mnmA* gene deletion line in *bsmnmA^+* parental background. Available from the Prigge lab |
| Genetic reagent (*P. falciparum*) | bsmnmA-yrvO^+ ΔmnmA | This paper | | *mnmA* gene deletion line in *bsmnmA-yrvO^+* parental background. Available from the Prigge lab |
| Genetic reagent (*P. falciparum*) | bsmnmA-yrvO^+ ΔsufS | This paper | | *sufS* gene deletion line in *bsmnmA-yrvO^+* parental background. Available from the Prigge lab |
| Chemical compound, drug | Racemic Mevalonolactone | MilliporeSigma, MO, USA | Catalogue no. (Cat#) M4667 | 50 μM |
| Chemical compound, drug | WR99210 | Jacobus Pharmaceuticals, NJ, USA | | 2.5 nM |

*Appendix 1 Continued on next page*

*Appendix 1 Continued*

| Reagent type (species) or resource | Designation | Source or reference | Identifiers | Additional information |
|---|---|---|---|---|
| Chemical compound, drug | DSM1 | BEI Resources, VA, USA | Cat# MRA-1161 | 1.5 µM |
| Chemical compound, drug | Blasticidin S HCl | Corning Inc, NY, USA | Cat# 30–100-RB | 1.25 or 2.5 µg/mL |
| Chemical compound, drug | Anhydrous tetracycline (aTc) | Cayman Chemical, MI, USA | Cat# 10009542 | 0.5 µM |
| Chemical compound, drug | *Shield1* | Aobious Inc, MA, USA | Cat# AOB1848 | 0.5 µM |
| Software, algorithm | Prism V8.4 | GraphPad Software, CA, USA | | Parasite growth assay data analysis |
| Software, algorithm | Volocity | PerkinElmer, MA, USA | | Deconvolution and colocalization analysis of microscopy images |
| Software, algorithm | MEGA11 | *Tamura et al., 2021* https://doi.org/10.1093/molbev/msab120 | | Multiple sequence alignment and phylogenetic analysis |
| Software, algorithm | Phyre2 | https://www.sbg.bio.ic.ac.uk/servers/phyre2 | | Homology model prediction server |
| Software, algorithm | UCSF ChimeraX (version: 1.3rc202111110135) | https://www.rbvi.ucsf.edu/chimerax | | Molecular visualization program for structural model and structural alignment |
| Antibody | Anti-mouse horseradish peroxidase (HRP)-conjugated antibody (Sheep polyclonal) | MilliporeSigma, MO, USA | Cat# GENA931 | Immunoblot (1:10,000) |
| Antibody | Anti-aldolase antibody (Mouse monoclonal) | David J. Sullivan, Johns Hopkins Bloomberg School of Public Health | | Immunoblot (1:25,000) |
| Antibody | Anti-ACP antibody (Rabbit polyclonal) | *Gallagher and Prigge, 2010* https://doi.org/10.1002/prot.22582 | | Immunofluorescence assays (1:500) |
| Antibody | Anti-mouse Alexa 488 (Goat polyclonal) | Thermo Fisher Scientific, MA, USA | Cat# A11029 | Immunofluorescence assays (1:1000) |
| Antibody | Anti-rabbit Alexa 594 (Goat polyclonal) | Thermo Fisher Scientific, MA, USA | Cat# A11037 | Immunofluorescence assays (1:1000) |
| Other | ProLong Gold 4',6-diamidino-2-phenylindole (DAPI) antifade reagent | Thermo Fisher Scientific, MA, USA | Cat# P36935 | Mountant, Immunofluorescence |
| Other | DAPI | Thermo Fisher Scientific, MA, USA | Cat# 62248 | DNA-specific stain, 1 µg/mL |
| Other | MitoTracker CMX-Ros | Thermo Fisher Scientific, MA, USA | Cat# M7512 | Mitochondrion-specific stain, 30 nM |
| Other | *Bm* MnmA | PiroplasmaDB (https://piroplasmadb.org) | BMR1_03g03155 | *Babesia microti* MnmA protein sequence |
| Other | *Ta* MnmA | PiroplasmaDB (https://piroplasmadb.org) | TA12620 | *Theileria annulata* MnmA protein sequence |
| Other | *Tg* MnmA | ToxoDB (https://toxodb.org) | TGME49_309110 | *Toxoplasma gondii* MnmA protein |
| Other | *Et* MnmA | ToxoDB (https://toxodb.org) | ETH_00027940 | *Eimeria tenella* MnmA protein sequence |
| Other | *Bs* MnmA | Uniprot (https://uniprot.org) | O35020 | *Bacillus subtilis* MnmA protein sequence |

*Appendix 1 Continued on next page*

*Appendix 1 Continued*

| Reagent type (species) or resource | Designation | Source or reference | Identifiers | Additional information |
|---|---|---|---|---|
| Other | *Bs* YrvO | Uniprot ([https://uniprot.org](https://uniprot.org)) | O34599 | *B. subtilis* YrvO protein sequence |
| Other | *Bs* SufS | Uniprot ([https://uniprot.org](https://uniprot.org)) | O32164 | *B. subtilis* SufS protein sequence |
| Other | *Ec* SufE | Uniprot ([https://uniprot.org](https://uniprot.org)) | P76194 | *Escherichia coli* SufE protein sequence |
| Other | *Ec* MnmA | Uniprot ([https://uniprot.org](https://uniprot.org)) | P25745 | *E. coli* MnmA protein sequence |
| Other | *Ec* SufS | Uniprot ([https://uniprot.org](https://uniprot.org)) | P77444 | *E. coli* SufS protein sequence |
| Other | *Ec* IscS | Uniprot ([https://uniprot.org](https://uniprot.org)) | P0A6B7 | *E. coli* IscS protein sequence |
| Other | *Sy* MnmA | Uniprot ([https://uniprot.org](https://uniprot.org)) | Q2JY34 | *Synechococcus* sp. MnmA protein sequence |
| Other | *At* MnmA | Uniprot ([https://uniprot.org](https://uniprot.org)) | Q9SYD2 | *Arabidopsis thaliana* MnmA protein sequence |
| Other | *At* cNif1 | Uniprot ([https://uniprot.org](https://uniprot.org)) | Q93WX6 | Protein sequence of *A. thaliana* Nif1 protein in the chloroplast |
| Other | *At* mNif1 | Uniprot ([https://uniprot.org](https://uniprot.org)) | O49543 | Protein sequence of *A. thaliana* Nif1 protein in the mitochondrion |
| Other | *Sc* Mtu1 | Uniprot ([https://uniprot.org](https://uniprot.org)) | Q12093 | *Saccharomyces cerevisiae* Mtu1 protein sequence |
| Other | *Sc* Nfs1 | Uniprot ([https://uniprot.org](https://uniprot.org)) | P25374 | *S. cerevisiae* Nfs1 protein sequence |

