## [Editor Report]

This study provides important new insights into iron sulfur biosynthesis in the human malaria parasite *Plasmodium falciparum*. The work is based on elegant and robust genetic approaches, and not only confirms the essentiality of the plastid-hosted Suf iron-sulfur cluster synthesis pathway, but also highlights an important additional role for the cysteine desulfurase SufS in apicoplast maintenance via tRNA modification. The work provides compelling evidence for a dual function of parasite SufS, although impact on tRNA has not been established directly. These findings reveal a potential new target for metabolic intervention and will be of interest to researchers studying apicomplexan parasites, and more broadly, in the field of plastid biology.

---

## [Decision Letter]

**Decision letter after peer review:**

Thank you for submitting your article "The *Plasmodium falciparum* apicoplast cysteine desulfurase provides sulfur for both iron sulfur cluster assembly and tRNA modification" for consideration by *eLife*. Your article has been reviewed by 3 peer reviewers, one of whom is a member of our Board of Reviewing Editors, and the evaluation has been overseen by Anna Akhmanova as the Senior Editor. The reviewers have opted to remain anonymous.

Essential revisions:

All three reviewers appreciated that this is an important and elegant study that substantially advances our understanding of the metabolic functions of Plasmodium apicoplast. The study as it stands provides compelling evidence that SufS plays a dual role in the FeS synthesis pathway and apicoplast maintenance. However, there are a few points that should be addressed in a revised version of the manuscript:

1) The claim that SufS role in apicoplast maintenance is linked with tRNA modification is inferred from the genetic complementation experiments and the assumption that SufS plays a similar enzymatic role as the bacterial enzymes. However, this is not directly supported by data. In the absence of analysis of tRNA thiolation levels in the parasite (which would strengthen the study and make it complete), the authors should modify the text (and title) to clarify that the effect on tRNA remains a hypothesis that will require confirmation in follow-up work.

2) A more detailed phylogenetic analysis and sequence comparisons not only of MnmA but also of SufS should be included (see comments from reviewers #1 and #2).

3) The authors need to clarify the experimental conditions for the experiments shown in Figure 5F and 6F (see reviewer 1 point #2), if necessary through additional experiments.

Further suggestions to improve the manuscript are listed below in the reviewers' recommendations for the authors.

*Reviewer #1 (Recommendations for the authors):*

1) This work is elegant and robust. However, the claim that SufS (via MnmA) mediates tRNA modification remains a hypothesis but is not supported by data. The observation that BsYrvO and BsMnmA can complement the function of PfSufS and PfMnmA, respectively, strongly suggests that SufS plays a similar enzymatic role in *P. falciparum* as YrvO in Bacillus (i.e. sulfur transfer to MnmA for tRNA thiolation). However this is not formally demonstrated, and the impact on apicoplast integrity could reflect other functions of the protein. Experiments directly showing the alteration of apicoplast tRNA thiolation would make the story even more complete. Furthermore, the authors show comparisons between Pf and Bs MnmA, but it would be useful to also include information on the sequence/structural similarities between SufS and YrvO. In particular, what are the common features of SufS and YrvO that would not be observed in other bacterial cysteine desulfurases (*E. coli*…), which do not transfer sulfur to MnmA?

2) In the genetic complementation experiments in Figure 5F and 6F, the authors show that under non-permissive conditions (at day 8) there is disruption of the apicoplast. However, these experiments were apparently performed in the absence of mevalonate rescue, in conditions where a severe growth defect is observed. Therefore the effect on the apicoplast could be related to the parasite death. A more convincing approach would be to document the disruption of the apicoplast in a well-growing parasite in the presence of mevalonate. This would strengthen the conclusion that BsMnmA complements PfMnmA specific function in apicoplast maintenance.

3) In Figure 3F, how do the authors explain that knockdown has no effect on growth during the 4 first days, despite efficient protein depletion as shown in Figure S3A? Is it due to the inertia of the system for metabolite depletion?

4) While the paper is very well written with a clear narrative, some aspects of the presentation of data or experiments could be improved:

– In Figure 1D, it would be useful to show a normal apicoplast in the parental line

– In Fig3E, the apicoplast looks different (smaller) than in other parasites, is it representative of the normal apicoplast morphology?

– Line 338-340: The rationale for the experiment in Figure 7 could be explained in a more explicit way. The experiment is designed to test if YrvO is capable of complementing PfSufS functions, especially for apicoplast maintenance.

*Reviewer #2 (Recommendations for the authors):*

This is a very complete study, presenting an already considerable amount of data that adequately support the conclusions drawn by the authors. However, I feel that the tRNA modification part of the work (which is the most original part of the study) could have been explored further. The complementation assays with the bacterial BsMnmA strongly suggest that PfSufS would be able to directly transfer sulfur to the bacterial protein, however, tRNA analysis in the PfSufS and PfMnmA (and its complemented cell line) would provide compelling evidence that the key impact of depleting these proteins is indeed on thiolation of tRNAs. tRNAs can be purified and then identified and quantified by performing size exclusion chromatography and mass spectrometry analyses (see PubmedID 30287681 for a recent example in Plasmodium).

In the same line of thoughts, several bacterial Mnma seem to show strong specificity to their partner desulfurase and in some bacterial lineages there is even indirect sulfur transfer to the tRNA via sulfur-carrier protein, so it is in fact quite unexpected to see that PfSufS could be a direct sulfur donor to a procaryotic MnmA, which in turn is potentially able to perform sulfur transfer directly on the parasite tRNA (which is, as mentioned above, not demonstrated here). Recent work has shown that some bacterial sulfur transferases may contain specific features linked to the ability to directly incorporate sulfur into tRNAs (see PubmedID 31801798). Given the cyanobacterial origin of the apicoplast, would it be possible that PfMnma is more closely related to procaryotic Mnma enzymes that are able to perform direct sulfur transfer? To complement the protein sequence alignments currently provided in Figure 3 supplements 1&2, it may be quite informative to include a small phylogenetic analysis coupled with a comparison of the sequence features of apicomplexan MnmA with procaryotic MnmA (including cyanobacterial sequences).

Finally, in complement to the data presented on Figure 7, depleting PfSufS in the bsmnmA+ (thus not expressing the YrvO desulfurase) cell line is likely to lead to both a loss of the apicoplast and a loss of parasite viability. This would be strengthening the idea that SufS acts by transferring sulfur to BsMnmA and would rule out an unrelated effect of MnmA alone.

*Reviewer #3 (Recommendations for the authors):*

1) The authors could elaborate some more on the result of the Mev-dependent phenotype of δ mnmA construct (Figure 3L). This genetic construct presumably would contain an intact Fe-S biosynthetic pathway; however, loss of the apicoplast genome encoding sufB, ultimately leads to defects in Fe-S biogenesis, explaining the result.

2) The authors commented on the experimental challenge of generating a bsmnmA+ on a deltamnmA parasite line. Did the authors sequence the sufS gene in this resulting construct to rule out the possibility of selecting a sufS mutant that enabled this complementation? Along the same lines, could the authors comment on the expression levels of the Bacillus' enzymes when compared to the endogenous levels of PfSufS and PfMnmA?

3) Although this reviewer favors the conclusions of this study on the proposed role of PfSufS and PfMnmA in 2-thiouridine tRNA modification and the genetic experiments supporting this role, the results described in this study do not provide a direct demonstration of their involvement or exclusive involvement in tRNA thiolation. That is, no analysis of tRNA modifications was included in this report. I'm certainly sensitive to the experimental challenges associated with these experiments and the complexity of analyzing any data resulting from samples of an organelle of an intracellular parasite. The purpose of this comment is not to invalidate the value of this contribution or to set an expectation for additional experiments, as the report is already describing several novel findings. But an ask is to revise the document to indicate that instead of "establishing" a role in tRNA thiolation, the results support this proposed role. Also, perhaps adding a comment in the discussion would add value to readers being able to appreciate the complexity of multiple pathways involving 2sU in the organism and host.

---

## [Author Response]

Essential revisions:All three reviewers appreciated that this is an important and elegant study that substantially advances our understanding of the metabolic functions of Plasmodium apicoplast. The study as it stands provides compelling evidence that SufS plays a dual role in the FeS synthesis pathway and apicoplast maintenance. However, there are a few points that should be addressed in a revised version of the manuscript:1) The claim that SufS role in apicoplast maintenance is linked with tRNA modification is inferred from the genetic complementation experiments and the assumption that SufS plays a similar enzymatic role as the bacterial enzymes. However, this is not directly supported by data. In the absence of analysis of tRNA thiolation levels in the parasite (which would strengthen the study and make it complete), the authors should modify the text (and title) to clarify that the effect on tRNA remains a hypothesis that will require confirmation in follow-up work.

We modified the text throughout the manuscript to address this issue. These changes and an explanation for why we didn’t measure tRNA thiolation levels are described in more detail in the responses to the reviewers below. A basic conceptual problem with measuring changes in tRNA thiolation levels in the SufS or MnmA KO parasites is the loss of the apicoplast genome and loss of all apicoplast tRNAs.

2) A more detailed phylogenetic analysis and sequence comparisons not only of MnmA but also of SufS should be included (see comments from reviewers #1 and #2).

We performed the suggested phylogenetic and sequence analyses and made some interesting observations that may impact how MnmA receives sulfur. Please, see our response to Reviewer #1, second part of point #1.

3) The authors need to clarify the experimental conditions for the experiments shown in Figure 5F and 6F (see reviewer 1 point #2), if necessary through additional experiments.Further suggestions to improve the manuscript are listed below in the reviewers' recommendations for the authors.

We clarified our experimental conditions. Please, see our detailed response to reviewer #1, point #2.

Reviewer #1 (Recommendations for the authors):1) This work is elegant and robust. However, the claim that SufS (via MnmA) mediates tRNA modification remains a hypothesis but is not supported by data. The observation that BsYrvO and BsMnmA can complement the function of PfSufS and PfMnmA, respectively, strongly suggests that SufS plays a similar enzymatic role in *P. falciparum* as YrvO in Bacillus (i.e. sulfur transfer to MnmA for tRNA thiolation). However this is not formally demonstrated, and the impact on apicoplast integrity could reflect other functions of the protein. Experiments directly showing the alteration of apicoplast tRNA thiolation would make the story even more complete.

We appreciate the reviewer’s enthusiasm for our findings. The reviewer agreed that our complementation experiments of PfSufS and PfMnmA activity with biochemically and genetically well-characterized bacterial orthologs strongly suggest dual function of PfSufS: both in FeS cluster formation and tRNA modification. As the reviewer points out, we did not measure tRNA modifications directly to show that they are affected by the loss of PfSufS and/or PfMnmA. We thought a lot about how to measure tRNA modifications, but concluded that there are significant technical limitations:

i) Both the apicoplast and the nuclear genomes of *P. falciparum* express tRNALysUUU, tRNAGluUUC, and tRNAGlnUUG. In the apicoplast, MnmA should produce 2-thioU at residue 34 of these tRNAs. In the cytosol, an enzyme that does not have homology to MnmA (NCS2) is predicted to produce the same 2-thioU modification at the same residue of the three tRNAs expressed by the nuclear genome. Current methods of detecting tRNA base modifications involve hydrolyzing tRNA pools followed by LC-MS detection of modified bases. Thus, we would likely have to devise methods to separate apicoplast tRNAs from nuclear tRNAs prior to conducting an analysis of tRNA base modifications.

ii) Low levels of apicoplast tRNAs would complicate efforts to study base modifications. Microarray analysis of apicoplast tRNAs shows that they are expressed at low levels compared to their nuclear counterparts (PMID: 32059044). Thus, we would probably also have to increase the parasite mass significantly to match the amount of tRNAs used in the previously described analysis of *P. falciparum* tRNA base modifications (PMID: 30287681). In that study, the authors showed that total apicoplast RNA is a relatively minor population (rRNAs 0.5–2% of nuclear-encoded rRNAs). Due to the relatively poor contribution of the apicoplast RNAs, the authors speculated that most modifications from this organelle would have been below the detection limit of the LC-MS method they employed.Assuming the issues with detecting apicoplast tRNA modifications could be resolved, it would be difficult to link deletion of SufS or MnmA to specific changes in tRNA modifications. This is because deletion of either SufS or MnmA causes loss of the apicoplast genome and loss of all apicoplast tRNAs.

iii) Assuming the issues with detecting apicoplast tRNA modifications could be resolved, it would be difficult to link deletion of SufS or MnmA to specific changes in tRNA modifications. This is because deletion of either SufS or MnmA causes loss of the apicoplast genome and loss of all apicoplast tRNAs.

iv) Another way to test the enzymatic role of PfMnmA in tRNA modification is expression and purification of a recombinant version of the protein and test in vitro activity on its substrate tRNAs. We set out to do this, however, we could not get the PfMnmA gene PCR amplified or synthesized due to its large size and extreme AT content despite multiple attempts. This prevented us from producing recombinant PfMnmA protein or conducting complementation experiments in bacterial systems.

In light of these technical challenges, we consider the direct measurement of apicoplast tRNA modifications to be a separate project that will have technical challenges and require the development of new methods. Instead of tackling these challenges in this manuscript, we instead softened the claims made in the text by replacing words such as ‘demonstrating’ and ‘showing’ with statements about how the data support a role in tRNA thiolation. For example, the abstract now reads (line 42-44):

“Complementation of MnmA and SufS activity with a bacterial MnmA and its cognate cysteine desulfurase, strongly suggests that the parasite SufS provides sulfur for both FeS biosynthesis and tRNA modification in the apicoplast.”

Furthermore, the authors show comparisons between Pf and Bs MnmA, but it would be useful to also include information on the sequence/structural similarities between SufS and YrvO. In particular, what are the common features of SufS and YrvO that would not be observed in other bacterial cysteine desulfurases (*E. coli*…), which do not transfer sulfur to MnmA?

As suggested by the reviewer, we have now performed multiple sequence alignment and phylogenetic analysis with several cysteine desulfurases including PfSufS and BsYrvO. These data are presented as Figure 7—figure supplement 1. Our analysis shows that there is no common feature that differentiates PfSufS and BsYrvO from other cysteine desulfurases that are known to use intermediate sulfur transferases for sulfur transfer to MnmA (or MnmA orthologs). In fact, our analysis shows that PfSufS and BsYrvO are members of two different classes of cysteine desulfurases. These results do not allow us to make any suggestions as to why these two would function in a similar manner. We added statements in the discussion considering our findings from this bioinformatic analysis (line 482-486):

“We could not identify any conserved motif(s) between *Pf* SufS and *Bs* YrvO which would be indicative of their ability to transfer sulfur directly to their respective MnmA proteins. In fact, we found that these proteins represent two different classes of cysteine desulfurase (Figure 7—figure supplement 1). Further studies are necessary to understand what allows *Pf* SufS (and *Bs* YrvO) able to transfer sulfur directly.”

We also performed a detailed sequence and structural analysis of PfMnmA and its orthologs from prokaryotes (including cyanobacteria) and eukaryotes. Homology based structure prediction of MnmA orthologs revealed a classical MnmA-like domain in residues 4221084 (*P. falciparum* numbering) region of PfMnmA and in other apicomplexan orthologs. Phylogenetic analysis of this domain, though, shows diversity among the MnmA orthologs in recent times and there is no clustering among the MnmA enzymes that receive sulfur through intermediate sulfur transferases. These results are now presented as Figure 3—figure supplement 3. Interestingly, homology modelling also predicted the possible presence of a SufE-like domain in the long N-terminal ends (residues 146-421, *P. falciparum* numbering) of apicomplexan MnmA orthologs including *P. falciparum*. This new result is shown as Figure 3—figure supplement 8. The SufE-like domain in the N-terminus of PfMnmA might facilitate the transfer of sulfur from PfSufS and explain why SufE (or other protein intermediates) are not required for this process. We included these new insights in the Discussion section (line 473477).

2) In the genetic complementation experiments in Figure 5F and 6F, the authors show that under non-permissive conditions (at day 8) there is disruption of the apicoplast. However, these experiments were apparently performed in the absence of mevalonate rescue, in conditions where a severe growth defect is observed. Therefore the effect on the apicoplast could be related to the parasite death. A more convincing approach would be to document the disruption of the apicoplast in a well-growing parasite in the presence of mevalonate. This would strengthen the conclusion that BsMnmA complements PfMnmA specific function in apicoplast maintenance.

The genetic complementation experiments shown in in 5F and 6F were performed in the absence of mevalonate rescue. The question here is whether loss of MnmA activity leads to parasite death and subsequent apicoplast disruption, or whether apicoplasts are disrupted in a way that can be rescued with mevalonate. To address this, we conducted a time-course of apicoplast disruption in the presence of mevalonate (as suggested). These new results are presented as Figure 3 —figure supplement 5, 6, and 7 and show that significant apicoplast loss occurs on day 4 of the 8-day experiment while parasite growth remains robust due to the presence of mevalonate. Since this question is now addressed in Figure 3 —figure supplement 5, 6, and 7, we did not modify Figures 5F or 6F.

3) In Figure 3F, how do the authors explain that knockdown has no effect on growth during the 4 first days, despite efficient protein depletion as shown in Figure S3A? Is it due to the inertia of the system for metabolite depletion?

In Figure 3F, we used the TetR-DOZI inducible knockdown tool to control expression of endogenous *Pf* MnmA. The slow kinetics of growth arrest shown in Figure 3F combined with the more rapid reduction in protein levels shown in current Figure 3-5A are not unusual, in our experience. We observed very similar results using the same knockdown method targeting the apicoplast ubiquitin-like protein (AUBL) (PMID: 32817449). There may be a ‘delayed death’ phenomenon at work since we see that the apicoplast organelles are intact on day 2 of the knockdown time course, but there is a significant increase in apicoplast disruption observed by day 4 (Figure 3 —figure supplement 6 and 7).

4) While the paper is very well written with a clear narrative, some aspects of the presentation of data or experiments could be improved:– In Figure 1D, it would be useful to show a normal apicoplast in the parental line

The suggested images of the fluorescently labeled apicoplast in the PfMev parental line have already been published previously in the article “A mevalonate bypass system facilitates elucidation of plastid biology in malaria parasites”, PMID: 32059044. We would have included these images, but since they have already been published and can be referenced elsewhere, we chose not to include them.

– In Fig3E, the apicoplast looks different (smaller) than in other parasites, is it representative of the normal apicoplast morphology?

The data presented in Figure 3E are from an immunofluorescence labelling assay (IFA) where the cells were fixed and permeabilized, before labelling with specific antibodies. All the other microscopy images presented in this manuscript are from live cell fluorescence imaging. In general, 5-15% reduction in cell length have been observed when treated with various commonly used fixatives in IFAs (PMID: 33912815), which could explain the difference in appearance of the apicoplast in Figure 3E from other images presented in this manuscript.

– Line 338-340: The rationale for the experiment in Figure 7 could be explained in a more explicit way. The experiment is designed to test if YrvO is capable of complementing PfSufS functions, especially for apicoplast maintenance.

We have clarified the rationale for the experiment presented in Figure 7 in the revised manuscript (line 344-352):

“In these parasites, we attempted to delete *sufS* with continuous supplementation of mevalonate to determine whether the *Bs* MnmA*-*YrvO protein can complement the essential activity of SufS. We tested this because not only have *Bs* MnmA and *Bs* YrvO been functionally characterized (Black & Santos, 2015a), but because *Pf* SufS and *Bs* YrvO share conserved catalytic residues required for cysteine desulfurase activity (Figure 7—figure supplement 1). We expected successful complementation of *Pf* SufS with *Bs* MnmA*-*YrvO to result in a parasite line with intact apicoplasts and a mevalonate-dependent phenotype (Figure 7A), which would suggest that sulfur acquired by *Bs* YrvO is only transferred to MnmA but not to the components of the parasite SUF pathway.”

Reviewer #2 (Recommendations for the authors):This is a very complete study, presenting an already considerable amount of data that adequately support the conclusions drawn by the authors. However, I feel that the tRNA modification part of the work (which is the most original part of the study) could have been explored further. The complementation assays with the bacterial BsMnmA strongly suggest that PfSufS would be able to directly transfer sulfur to the bacterial protein, however, tRNA analysis in the PfSufS and PfMnmA (and its complemented cell line) would provide compelling evidence that the key impact of depleting these proteins is indeed on thiolation of tRNAs. tRNAs can be purified and then identified and quantified by performing size exclusion chromatography and mass spectrometry analyses (see PubmedID 30287681 for a recent example in Plasmodium).

The suggestion here is to measure tRNA thiolation in the knockout and complemented parasite lines to link SufS and/or MnmA to tRNA modification. We addressed this point as described above for point 1 of reviewer 1 by modifying the strength of the claims made in the paper. As described above, it would be difficult using current methods to measure apicoplast tRNA modifications versus nuclear tRNA modifications. The same thiolated base is predicted to be independently produced in different compartments of the cell. There is also a conceptual problem with analyzing tRNA modifications in SufS or MnmA deletion lines since these lines lose the apicoplast genome and do not contain any apicoplast tRNAs.

In the same line of thoughts, several bacterial Mnma seem to show strong specificity to their partner desulfurase and in some bacterial lineages there is even indirect sulfur transfer to the tRNA via sulfur-carrier protein, so it is in fact quite unexpected to see that PfSufS could be a direct sulfur donor to a procaryotic MnmA, which in turn is potentially able to perform sulfur transfer directly on the parasite tRNA (which is, as mentioned above, not demonstrated here). Recent work has shown that some bacterial sulfur transferases may contain specific features linked to the ability to directly incorporate sulfur into tRNAs (see PubmedID 31801798). Given the cyanobacterial origin of the apicoplast, would it be possible that PfMnma is more closely related to procaryotic Mnma enzymes that are able to perform direct sulfur transfer? To complement the protein sequence alignments currently provided in Figure 3 supplements 1&2, it may be quite informative to include a small phylogenetic analysis coupled with a comparison of the sequence features of apicomplexan MnmA with procaryotic MnmA (including cyanobacterial sequences).

We thank the reviewer for the suggestion to include a sequence and phylogenetic analysis to compare MnmA orthologs. We analyzed MnmA sequences across kingdoms (including prokaryotic species) and specifically within apicomplexan species as well as sequence alignments that focus individually on the N-terminal and C-terminal domains. We also included an analysis of cysteine desulfurase enzymes and were able to add several interesting insights to the manuscript. Please see the response above to a similar suggestion from reviewer 1.

Finally, in complement to the data presented on Figure 7, depleting PfSufS in the bsmnmA+ (thus not expressing the YrvO desulfurase) cell line is likely to lead to both a loss of the apicoplast and a loss of parasite viability. This would be strengthening the idea that SufS acts by transferring sulfur to BsMnmA and would rule out an unrelated effect of MnmA alone.

The suggestion here is to generate a *bsmnmA*^+^Δ*sufS* parasite line to show that MnmA cannot compensate in some way for the loss of SufS. By ‘unrelated effect of MnmA alone’ we assume the reviewer is interested in whether MnmA+BsMnmA might be able to catalyze the liberation of sulfur from cysteine. If this was the case, we would have observed an intact apicoplast and mevalonate-independent growth phenotype in the Δ*sufS* line.

Reviewer #3 (Recommendations for the authors):1) The authors could elaborate some more on the result of the Mev-dependent phenotype of δ mnmA construct (Figure 3L). This genetic construct presumably would contain an intact Fe-S biosynthetic pathway; however, loss of the apicoplast genome encoding sufB, ultimately leads to defects in Fe-S biogenesis, explaining the result.

We thank the reviewer for this suggestion. We added a statement elaborating the effect of the loss of sufB in the MnmA knockout line (line 250-253):

“Additionally, ∆*mnmA* parasites were dependent on exogenous mevalonate supplementation for survival (Figure 3L), which can be explained by the loss of *sufB* resulting in defective FeS synthesis and the inability to synthesize isoprenoids (Akuh et al., 2022; Elahi & Prigge, 2023; Swift et al., 2022).”

2) The authors commented on the experimental challenge of generating a bsmnmA+ on a deltamnmA parasite line. Did the authors sequence the sufS gene in this resulting construct to rule out the possibility of selecting a sufS mutant that enabled this complementation? Along the same lines, could the authors comment on the expression levels of the Bacillus' enzymes when compared to the endogenous levels of PfSufS and PfMnmA?

The suggestion here was to sequence the *bs*mnmA^+^ ∆mnmA line to investigate the possibility of selecting a SufS mutant. We have now sequenced the *sufS* gene sequence from two clonal lines of the *bs*mnmA^+^ ∆*mnmA* parasites. We also sequenced *sufS* gene from one of the *bs*mnmA-yrvO^+^ ∆*mnmA* lines and the PfMev^attB^ parental line. We did not observe any mutation in the clonal lines of the *bs*mnmA^+^ ∆*mnmA* or in the *bs*mnmA-yrvO^+^ ∆*mnmA* parasites.

We added this finding in the Discussion section (line 437-439).

In both the *bs*mnmA^+^ and *bs*mnmA-*yrvO*^+^ lines, the *Bacillus* genes were expressed using the promoter for calmodulin, which is expressed at a level greater than 90% of other *P. falciparum* genes (and much higher than *mnmA* or *sufS*). We intentionally tried to produce the *Bacillus* proteins at high levels to facilitate imaging of the mCherry fusion proteins. Live fluorescence is difficult to detect in poorly expressed proteins and our imaging results suggest that the *Bacillus* proteins are produced at a high level.

3) Although this reviewer favors the conclusions of this study on the proposed role of PfSufS and PfMnmA in 2-thiouridine tRNA modification and the genetic experiments supporting this role, the results described in this study do not provide a direct demonstration of their involvement or exclusive involvement in tRNA thiolation. That is, no analysis of tRNA modifications was included in this report. I'm certainly sensitive to the experimental challenges associated with these experiments and the complexity of analyzing any data resulting from samples of an organelle of an intracellular parasite. The purpose of this comment is not to invalidate the value of this contribution or to set an expectation for additional experiments, as the report is already describing several novel findings. But an ask is to revise the document to indicate that instead of "establishing" a role in tRNA thiolation, the results support this proposed role. Also, perhaps adding a comment in the discussion would add value to readers being able to appreciate the complexity of multiple pathways involving 2sU in the organism and host.

We appreciate the reviewer’s consideration regarding the technical challenges of analyzing apicoplast tRNAs given their low expression and the presence of the same tRNA modifications in the apicoplast and nucleus. We discussed these challenges in response to a similar comment made by reviewer 1 (point 1). We followed the reviewer’s suggestion and softened our claims by replacing words such as ‘demonstrating’ and ‘showing’ with statements about how the data support a role in tRNA thiolation. For example, the abstract now reads (line 42-44):

“Complementation of MnmA and Sufis activity with a bacterial MnmA and its cognate cysteine desulfurase, strongly suggests that the parasite SufS provides sulfur for both FeS biosynthesis and tRNA modification in the apicoplast.”

As suggested, we added a sentence to the discussion to name the two enzymes that are predicted to catalyze s^2^U modifications in *P. falciparum* parasites (line 420-422). It is also true that host white blood cells also have the ability to form s^2^U tRNA modifications and that these cells are present in our culture system unless rigorous leukodepletion methods are used.

“In *P. falciparum*, NCS2 (PF3D7_1441000) is predicted to catalyze s^2^U modification of nuclear tRNAs while MnmA is predicted to make this modification in the apicoplast.”